# COLLIE: Guiding Skill Discovery in Semantically Coherent Latent Space

Yao Luan [* 1]  Ni Mu [* 1]  Hanfei Ge [2]  Yiqin Yang [3]  Bo Xu [3]  Qing-Shan Jia [1 4 5]

## Abstract

Unsupervised skill discovery (USD) aims to learn diverse behaviors without reward functions, but often results in task-irrelevant or hazardous behaviors due to uniform exploration. Guided skill discovery (GSD) addresses this issue by incorporating human intent to focus exploration on meaningful regions. However, existing GSD methods typically require training additional guidance models, and rely on pre-defined rules or expert demonstration, which can be ineffective under sparse, online-collected human feedback. To overcome this, we propose COLLIE, a GSD framework that leverages dense unsupervised data to construct a semantically coherent skill latent space. This latent space is well-structured, enabling reliable guidance with sparse online feedback. Moreover, its semantic coherence property enables training-free construction of guidance signals, eliminating the need for additional model training beyond skill learning. Theoretical analysis justifies the effectiveness of our training-free guidance signal, while experiments across diverse state-based and pixel-based tasks show that COLLIE learns diverse, human-aligned skills, avoids hazardous behaviors, and achieves superior downstream performance with minimal human feedback. Code is available at https://github.com/iiiiiii11/COLLIE.

---

[*]Equal contribution  [1]The Center for Intelligent and Networked Systems (CFINS), Department of Automation, Beijing National Research Center for Information Science and Technology, Tsinghua University, Beijing, China [2]Kuang Yaming College, Nanjing University, Nanjing, China [3]The Key Laboratory of Cognition and Decision Intelligence for Complex Systems, Institute of Automation, Chinese Academy of Sciences, Beijing, China [4]Beijing Key Laboratory of Embodied Intelligence Systems, Beijing, China [5]Institute for Embodied Intelligence and Robotics, Tsinghua University, Beijing, China. Correspondence to: Yiqin Yang <yiqin.yang@ia.ac.cn>, Qing-Shan Jia <jiaqs@tsinghua.edu.cn>.

*Proceedings of the 43rd International Conference on Machine Learning*, Seoul, South Korea. PMLR 306, 2026. Copyright 2026 by the author(s).

## 1. Introduction

Unsupervised learning aims to learn meaningful representations or behaviors through self-supervised objectives without pre-defined task-specific goals, which has shown effectiveness across domains, such as computer vision (Chen et al., 2020; Radford et al., 2021) and natural language processing (Devlin et al., 2019; Brown et al., 2020). In reinforcement learning, *unsupervised skill discovery* (USD) builds on this idea to learn diverse, distinguishable behaviors that broadly cover the state space, facilitating downstream tasks. However, USD's uniform exploration strategy often leads to useless or harmful skills (Kim et al., 2023), especially in complex scenarios, where vast state spaces include irrelevant or hazardous regions. This inefficiency not only wastes computational resources but also limits USD's practical applicability in real-world tasks.

To address the limitations of USD, *guided skill discovery* (GSD) methods draw inspiration from human cognition, where humans prioritize exploring potentially useful regions, rather than uniformly covering the state space (Du et al., 2023). By incorporating external human intent, GSD focuses exploration on meaningful and safe areas. However, existing GSD methods often ① rely on pre-defined rules or expert demonstrations (Kim et al., 2023; 2024), which can be challenging to obtain in complex environments, and ② require training auxiliary models to encode human intent (Klemsdal et al., 2021; Kim et al., 2024), which risk overfitting with limited human feedback, leading to unreliable guidance in complex scenarios, especially when expert knowledges are unavailable in advance.

To overcome these issues, we propose **CO**herent **L**atent-based guided ski**L**l d**I**scov**E**ry (COLLIE), a novel GSD method that leverages dense unsupervised data to construct a *semantically coherent* skill latent space for effective guidance. In this space, nearby embeddings correspond to states with similar human desirability. As this property is derived from dense unlabeled data, the latent space exhibits structural information, which complements sparse human feedback and ensures the effectiveness of the guidance signal in scenarios with only non-expert datasets. Additionally, this coherence allows COLLIE to generate a training-free guidance signal $w(s)$ by propagating semantics from a small set of labeled "good" or "bad" states, thereby avoiding ad-

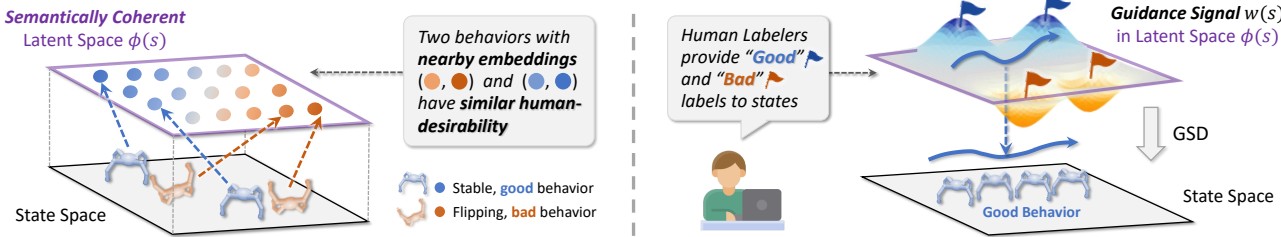

*Figure 1.* Overview of COLLIE. (1) We leverage dense unsupervised data to construct a *semantically coherent* latent space, where states with nearby embeddings share similar human desirability. (2) Using sparse human "good/bad" labels on states, COLLIE constructs a dense, training-free guidance signal $w(s)$.

ditional model training. To preserve the guidance signal's accuracy, which requires labeled states to sufficiently cover the state space, COLLIE employs an active query strategy to select under-explored behaviors for human labeling. By integrating the guidance signal into the intrinsic reward of USD, COLLIE directs exploration toward desirable regions, without relying on expert data or auxiliary model training.

Both theoretical analysis and experimental results validate the effectiveness of COLLIE. Theoretically, COLLIE enables effective, training-free guidance signal construction. Empirically, across various complex tasks, COLLIE learns diverse, human-aligned skills with minimal feedback, avoiding hazards, focusing on task-relevant regions, and enhancing downstream performance. Visualizations of learned skills further confirm that COLLIE enables meaningful exploration. In summary, our contributions are threefold:

- We propose COLLIE, a novel GSD framework that leverages dense unlabeled data to complement sparse human feedback. It guides skill learning within a semantically coherent latent space, thus preventing overfitting when expert demonstrations are unavailable.

- The semantic coherence property supports a training-free GSD paradigm with theoretical guarantees, which eliminates the need for auxiliary guidance models and ensures effective guidance with sparse feedback.

- Extensive experiments on both state-based and pixel-based tasks show that COLLIE learns diverse, safe, and human-aligned skills with minimal human feedback, while achieving downstream task performance approaching oracle-level guidance, demonstrating its effectiveness.

## 2. Preliminaries

**Unsupervised Skill Discovery (USD).** Unsupervised reinforcement learning considers a Markov decision process (MDP) without reward functions, which is characterized by the tuple $(\mathcal{S}, \mathcal{A}, \mathcal{P}, \mu_0)$. Here, $\mathcal{S}$ and $\mathcal{A}$ are the state and ac-

tion spaces, $\mathcal{P} : \mathcal{S} \times \mathcal{A} \rightarrow \Delta(S)$ denotes the state transition, and $\mu_0$ is the initial state distribution.

USD methods aim to acquire knowledge about the environment by learning a set of distinguishable skills that collectively cover the state space. This is achieved by introducing a latent skill space $\mathcal{Z}$ and training a skill-conditioned policy $\pi : \mathcal{S} \times \mathcal{Z} \rightarrow \Delta(\mathcal{A})$. During training, skills are sampled from a prior distribution $p(z)$, and the agent interacts with the environment using the policy $\pi(a|s, z)$ based on the sampled skill. Once learned, these skills can be reused to facilitate downstream tasks, such as (1) learning a hierarchical policy that uses skill-conditioned policies as lower-level policies, or (2) selecting skills that maximize the task reward in a zero-shot manner.

**Distance-maximizing Skill Discovery (DSD).** A common approach to USD is to maximize the mutual information (MI) $I(s, z)$ between skills $z$ and the visited states $s$ (Eysenbach et al., 2019). However, this can lead to static behaviors, as MI only ensures skill discriminability without encouraging broader state coverage (Park et al., 2024). DSD methods (Park et al., 2023; 2022b; 2024) address this by aligning state space distances to latent skill space distances. A typical objective is:

$$\sup_{\pi, \phi} \mathbb{E}_{\tau, z} \left[ \sum_{t=0}^{T-1} (\phi(s_{t+1}) - \phi(s_t))^\top z \right] \tag{1}$$
$$\text{s.t. } \|\phi(x) - \phi(y)\|_2 \le d(x, y), \ \forall x, y \in \mathcal{S},$$

where $\tau = (s_0, \ldots, s_{T-1})$ denotes the trajectory, $d(\cdot, \cdot)$ is a distance metric in the state space. This objective can be optimized via dual gradient descent (Boyd & Vandenberghe, 2004) with a Lagrange multiplier $\lambda$, and the policy is updated with intrinsic reward $r(s, z, s') = (\phi(s') - \phi(s))^\top z$.

**Guided Skill Discovery (GSD).** USD can be inefficient in practical scenarios, as many of the learned skills may be irrelevant or even harmful to downstream tasks (Kim et al., 2024). GSD addresses this issue by incorporating human intent to guide skill learning toward desirable behaviors

---

**Algorithm 1** COLLIE

---

**Require:** Feedback frequency $K$, total feedback number $N_{\text{total}}$, number of queries per feedback session $M$, total epoch number $T^{\text{e}}$

1: Initialize replay buffer $\mathcal{B}$, feedback buffer $\mathcal{D}_0, \mathcal{D}_1, \mathcal{D}_2$
2: **for** each epoch $e = 1, 2 \ldots, T^{\text{e}}$ **do**
3:     Sample skill $z \sim p(z)$, rollout with policy $\pi(a|s, z)$ and store $(s, a, s')$ into $\mathcal{B}$
4:     **if** epoch % $K = 0$ and $|\mathcal{D}_0| + |\mathcal{D}_1| + |\mathcal{D}_2| < N_{\text{total}}$ **then**
5:         Select segments $\{\sigma_i\}_{i=1}^M \sim \mathcal{B}$ using the method in Section 3.4 and Algorithm 2
6:         Query labelers for feedback $\{y_i\}_{i=1}^M$
7:         Save labeled states into feedback buffer, $\mathcal{D}_y \leftarrow \mathcal{D}_y \cup \{s : s \in \sigma_i, y_i = y\}_{i=1}^M, y = 0, 1, 2$
8:     **end if**
9:     Sample transitions from $\mathcal{B}$
10:    Calculate the guidance signal $w(s)$ with Eq. 7 and the smooth mechanism in Section 3.5
11:    Update the skill latent $\phi(s)$ with Eq. 11
12:    Update the Lagrange multiplier $\lambda$ with Eq. 12
13:    Update the policy $\pi(a|s, z)$ with Eq. 13
14: **end for**

---

while avoiding undesirable ones. A typical objective is:

$$\sup_{\pi, \phi} J_{\text{USD}}(\pi, \phi) + \lambda_{\text{guide}} \cdot J_{\text{guide}}(\pi, \phi)$$
$$\text{s.t.} \quad C_{\text{USD}}(\phi) \leq 0, \quad C_{\text{guide}}(\phi) \leq \lambda_{\text{guide}}^c, \tag{2}$$

where $J_{\text{USD}}(\pi, \phi)$ represents the USD objective, such as mutual information $I(s, z)$ or the DSD objective. $C_{\text{USD}}(\phi)$ denotes constraints in USD objectives, such as those in Eq. 1. $J_{\text{guide}}(\pi, \phi)$ reflects the guidance objective derived from human intent, which may include expert trajectories (Kim et al., 2024; Klemsdal et al., 2021) or pairwise human preferences (Hussonnois et al., 2023; 2025). $C_{\text{guide}}(\phi)$ is the guidance in the form of constraints, like analytical constraint formulas for safety (Kim et al., 2023). The coefficients $\lambda_{\text{guide}}, \lambda_{\text{guide}}^c \geq 0$ adjust the strength of guidance.

**Human Feedback Format.** We consider an interactive human-in-the-loop setting, where a labeler evaluates agent behaviors by providing feedback on state sequence segments $\sigma = (s_t, \ldots, s_{t+H-1})$ of fixed length $H$, also referred to as "queries" in this paper. Each segment is assigned a scalar label $y \in \{0, 1, 2\}$, where $y = 2$ denotes a "good" segment (e.g., moving toward a goal), $y = 0$ denotes a "bad" segment (e.g., entering a hazardous area), and $y = 1$ denotes a "neutral" segment (e.g., neither contributing to task goals nor incurring risk). We assume all states within a segment share the same label. Labeled states are stored in a dataset $\mathcal{D} = \{(s, y)\}$, which is partitioned into subsets $\mathcal{D}_0, \mathcal{D}_1$, and $\mathcal{D}_2$ for bad, neutral, and good states, respectively. Note that

this feedback format differs from recent preference-based RL methods. Further discussions on this choice are detailed in Appendix F.

## 3. COLLIE: Coherent Latent-based GSD

To address the ineffectiveness of training additional guidance models in GSD with sparse human feedback, we propose COLLIE, a coherent latent-based guided skill discovery method, as illustrated in Fig. 1 and Algorithm 1. COL-LIE ensures reliable guidance from limited feedback by leveraging a semantically coherent latent space. Specifically, Section 3.1 outlines the GSD framework. Built upon this framework, Section 3.2 defines the semantic coherence property required for the latent space $\phi(s)$ and describes its construction method. Section 3.3 then utilizes this property to derive the guidance signal $w(s)$ in a training-free manner.

### 3.1. GSD Framework With the Guidance Signal

To enable effective guidance under sparse feedback, we require a latent skill space trained with dense unsupervised data, which meaningfully reflects the structure of the state space. We therefore build upon the DSD framework (Section 2), which learns a latent space $\phi(s)$ constrained to reflect state-space distances, and learns skills to maximize the distance traveled within the latent space. To incorporate human intent, we introduce a guidance signal $w(s) : \mathcal{S} \rightarrow \mathbb{R}^+$ as a distance modifier. Formally, the DSD objective (Eq. 1) is extended as:

$$\sup_{\pi, \phi} \mathbb{E}_{\tau, z} \left[ \sum_{t=0}^{T-1} \left( \phi(s_{t+1}) - \phi(s_t) \right)^\top z \right]$$
$$\text{s.t.} \ \|\phi(x) - \phi(y)\|_2 \leq w(x)d(x, y), \ \forall x, y \in \mathcal{S}, \tag{3}$$

This formulation captures a key intuition: assigning large $w(s)$ to human-desirable states relaxes the constraint on the latent space $\phi$, allowing for broader exploration in those areas. Conversely, small $w(s)$ in undesirable regions tightens the constraint, discouraging exploration. In essence, if we can construct a $w(s)$ that aligns with human intent, this framework naturally leads to human-desirable skills, as validated in prior works (Kim et al., 2024). Consequently, the complexity of GSD is reduced to constructing an effective, training-free guidance signal $w(s)$, which we discuss below.

### 3.2. Semantically Coherent Latent Space

Having established the GSD framework, we now address the prerequisites for constructing the guidance signal $w(s)$ in a training-free manner. This requires the latent space $\phi(s)$ to be *semantically coherent*, i.e., nearby embeddings correspond to states with similar human desirability. Under this property, states labeled as "good" are mainly surrounded by other good states, while "bad" states are similarly sur-

rounded by other bad states. Consequently, sparse human-provided labels can be effectively propagated across the latent space, allowing us to infer a dense guidance signal $w(s)$ from a few labeled states, without requiring additional model training.

We formalize the concept of *semantic coherence* as follows. Let $g(s) : \mathcal{S} \to \{0, 1, 2\}$ denote the human desirability for state $s$, with higher values indicating more desirable states. For a latent space $\mathcal{U}$ defined by $u = \phi(s) : \mathcal{S} \to \mathcal{U}$, we say $\mathcal{U}$ is semantically coherent, if $\forall \epsilon > 0, \exists \delta > 0$, such that

$$\begin{aligned} \|\phi(s_1) - \phi(s_2)\|_2 \leq \delta \implies \\ P[g(s_1) = g(s_2)] \geq 1 - \epsilon \quad \forall s_1, s_2 \in \mathcal{S}. \end{aligned} \quad (4)$$

We would like to note that constructing a semantically coherent latent space is necessary, because the raw state space is typically not semantically coherent. For example, in robotic locomotion, a robot at an arbitrary position $(x, y)$ can be in either a stable (human-desirable) or fallen (undesirable) state. Although these states may be very close in the state space, e.g., differing only in joint angles or orientation, they exhibit completely different human desirability. In these scenarios, Euclidean distance becomes an inadequate measure of semantic similarity (Jiang et al., 2025).

As directly constructing a latent space that satisfies Eq. 4 is challenging, we instead turn to a surrogate, leveraging the observation that successive states along a trajectory share similar desirability (Park et al., 2022a; Wang et al., 2019; Hamedi & Shad, 2022). This observation can be formalized as $P[g(s) = g(s')] \geq 1 - \epsilon, \exists \epsilon > 0, \forall (s, s') \in \mathcal{S}_{\text{adj}}$, where $\mathcal{S}_{\text{adj}}$ denotes the set of adjacent state pairs within trajectories. Consequently, to promote semantic coherence, we constrain the embeddings of such adjacent states to remain close:

$$\|\phi(s') - \phi(s)\|_2 \leq \delta_0, \quad \forall (s, s') \in \mathcal{S}_{\text{adj}}, \quad (5)$$

where $\delta_0 > 0$ is a constant. Then, by replacing the constraint in Eq. 3 with $\|\phi(s') - \phi(s)\|_2 \leq \delta_0 w(s), \forall (s, s') \in \mathcal{S}_{\text{adj}}$, we can derive the required semantically coherent GSD framework. As shown in Park et al. (2024), this local constraint implies a global Lipschitz condition with respect to temporal distance (Kaelbling, 1993; Hartikainen et al., 2019; Durugkar et al., 2021): $\|\phi(s_1) - \phi(s_2)\| \leq \delta_0 d_{\text{temp}}(s_1, s_2), \forall s_1, s_2 \in \mathcal{S}$, where $d_{\text{temp}}(s_1, s_2)$ is the minimum number of steps to transition from $s_1$ to $s_2$. This result connects our semantically coherent approach to Park et al. (2024).

### 3.3. Training-free Guidance Signal Construction

Building upon the semantically coherent latent space, we now construct the guidance signal $w(s)$ in a training-free manner. The key idea is that for any state $s$, its human desirability can be inferred from its distances to labeled states

within the semantically coherent latent space. Specifically, given a small set of human-labeled states $\mathcal{D} = \mathcal{D}_0 \cup \mathcal{D}_1 \cup \mathcal{D}_2$, we compute the minimum L2-distance from $s$ to each label set in the latent space:

$$d_\phi(s, \mathcal{D}') = \min_{s_0 \in \mathcal{D}'} \|\phi(s_0) - \phi(s)\|, \quad \mathcal{D}' \in \{\mathcal{D}_i\}_{i=0}^2. \quad (6)$$

We then define $w(s)$ as a soft assignment over the three desirability levels:

$$w(s) = \text{softmax}\big([-d_\phi(s, \mathcal{D}_i)]_{i=0}^2\big)[0, 1, 2]^\top, \quad (7)$$

where $\text{softmax}([x_1, \ldots, x_n]) = \frac{[\exp(x_1), \ldots, \exp(x_n)]}{\sum_{i=1}^n \exp(x_i)}$. This formulation intuitively assigns higher values to states closer to "good" regions and lower values to those near "bad" regions, ensuring $w(s)$ accurately reflects relative desirability.

This construction is not only intuitive but also theoretically grounded. As formally established in Proposition 3.1 (proof in Appendix B), when the constructed $w(s)$ is interpreted as a classifier, its asymptotic error rate is bounded by twice the Bayes error rate. This result justifies the reliability of our training-free guidance signal $w(s)$.

**Proposition 3.1.** *Consider the induced classifier $\hat{g}(s)$ derived from the guidance signal $w(s)$, i.e., $\hat{g}(s) = \arg\max_k \frac{\exp(-d_\phi(s, \mathcal{D}_k))}{\sum_{j=0}^2 \exp(-d_\phi(s, \mathcal{D}_j))}$. The asymptotic expected error rate of the classifier $\hat{g}(s)$ is bounded by the Bayes error rate $P^*(s)$, which is formally expressed as:*

$$P(\hat{g}(s) \neq g(s)) \leq 2P^*(s) - \frac{3}{2}[P^*(s)]^2. \quad (8)$$

**Full Objective.** Combining them all, we replace the constraint in Eq. 3 with $\|\phi(s') - \phi(s)\|_2 \leq \delta_0 w(s), \forall (s, s') \in \mathcal{S}_{\text{adj}}$ to maintain semantic coherence of the latent skill space, setting $\delta_0 = 1$ for consistency with Park et al. (2024). The constructed guidance signal $w(s)$ serves as a weighting factor for the DSD objective, leading to the following objective:

$$\sup_{\pi, \phi} \mathbb{E}_{\tau, z} \left[ \sum_{t=0}^{T-1} (\phi(s_{t+1}) - \phi(s_t))^\top z \right] \quad (9)$$
$$\text{s.t. } \|\phi(s') - \phi(s)\|_2 \leq w(s), \ \forall (s, s') \in \mathcal{S}_{\text{adj}}.$$

However, directly optimizing Eq. 9 is challenging, as the guidance signal $w(s)$ is embedded in the latent space constraint, and directly impacts the update of the latent $\phi(s)$. This coupling leads to instability, especially since our $w(s)$ is updated dynamically with incoming human feedback. To address this, we follow Kim et al. (2024) and derive a nearly equivalent but more practical objective of Eq. 9:

$$\sup_{\pi, \phi} \mathbb{E}_{\tau, z} \left[ \sum_{t=0}^{T-1} w(s_t) (\phi(s_{t+1}) - \phi(s_t))^\top z \right] \quad (10)$$
$$\text{s.t. } \|\phi(s') - \phi(s)\|_2 \leq 1, \ \forall (s, s') \in \mathcal{S}_{\text{adj}}.$$

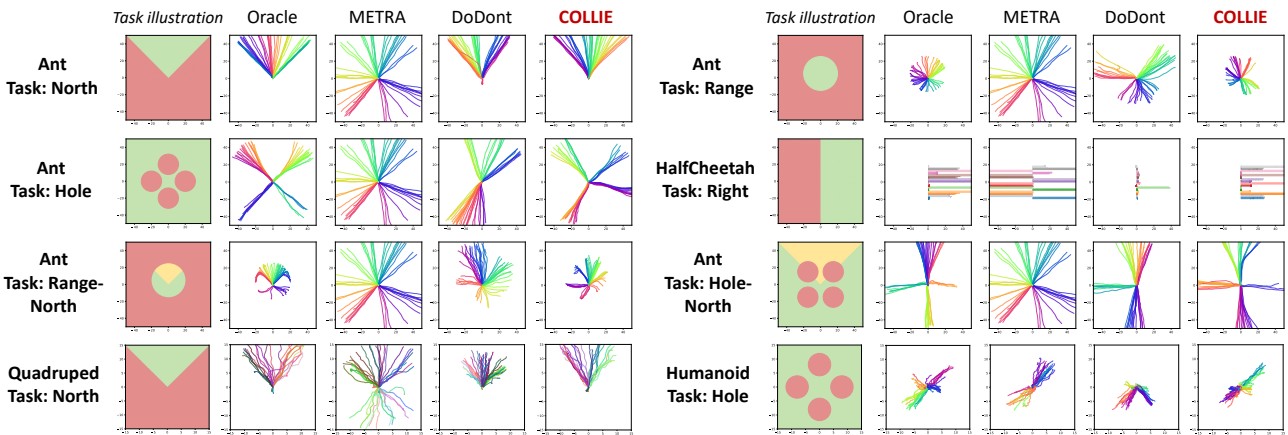

*Figure 2.* Visualizations of skills learned by COLLIE and baseline methods, by plotting x-y (or x) trajectories sampled from the learned policies. Different colors represent distinct skills $z$. In task illustrations, human-undesirable regions ($y = 0$) are highlighted in red, desirable regions ($y = 2$) in yellow, and neutral regions in green. COLLIE effectively aligns diverse skills with human intent.

Specifically, this derivation is based on a variable substitution, where we replace the latent function with a scaled version, $\phi'(s) = \phi(s)/w(s)$. Appendix C provides the formal derivation. This reformulation offers a crucial advantage: it decouples the guidance signal $w(s)$ from the DSD's latent space learning. It preserves the stability and latent space structure of the original DSD framework, while injecting human guidance by simply scaling DSD's intrinsic reward $r(s, z, s') = w(s)(\phi(s') - \phi(s))^\top z$.

Eq. 10 could be optimized by updating the latent $\phi$ and the Lagrange multiplier $\lambda$ to maximize Eq. 11 and minimize Eq. 12, and updating the policy $\pi$ to maximize the accumulated intrinsic reward in Eq. 13:

$$\mathcal{J}^\phi = \mathbb{E}_{(s,z,s')\sim\mathcal{B}}[w(s)(\phi(s') - \phi(s))^\top z$$
$$+ \lambda \min(\epsilon, 1 - \|\phi(s') - \phi(s)\|_2^2)] \quad (11)$$

$$\mathcal{J}^\lambda = \mathbb{E}_{(s,z,s')\sim\mathcal{B}}[\lambda \min(\epsilon, 1 - \|\phi(s') - \phi(s)\|_2^2)] \quad (12)$$

$$r(s, z, s') = w(s)(\phi(s') - \phi(s))^\top z \quad (13)$$

### 3.4. Active Query Strategy

The effectiveness of the training-free guidance signal, $w(s)$, relies on accurately estimating human desirability for a given state, via its nearest labeled neighbors in the latent space. This requires the labeled states set $\mathcal{D}$ to sufficiently cover the state space. To achieve this, we propose an active query strategy that prioritizes less-visited states by maximizing their state entropy in the labeled states $H_{\text{state}}(s) = -\log \Pr_{s\sim\mathcal{D}}(s)$. Since directly computing state entropy is intractable, we use an efficient particle-based entropy estimation (Singh et al., 2003; Liu & Abbeel, 2021b):

$$H_{\text{state}}(s) \approx \log\left(1 + \frac{1}{k}\sum_{j=1}^{k} \|s - s^{(j)}\|\right), \quad (14)$$

where $s^{(j)}$ denotes the $j$-th nearest neighbor of state $s$ in the labeled state dataset $\mathcal{D}$. We then define the query selection score $I(\sigma)$ for a segment $\sigma$ as $I(\sigma) = \sum_{s\in\sigma} H_{\text{state}}(s)$. Segments with higher $I(\sigma)$ are prioritized for human labeling. Further details on the query strategy are provided in Appendix A.

### 3.5. Implementation Details

**Algorithm outline.** Algorithm 1 and Fig. 1 outline the procedure of COLLIE. Building on the DSD framework, we iteratively collect feedback from the learned skills (lines 4~8) and immediately employ the feedback to construct the guidance signal (line 10), enabling the guidance signal to be updated online with the skills.

**Computational efficiency.** Although the computation of the guidance signal $w(s)$ involves calculating distances $d_\phi(s, \mathcal{D})$ over all labeled states, the computational burden is not heavy. This is because the number of labeled states is small, and we cache the embeddings $\phi$ of these states to further accelerate the process.

**Guidance signal smoothing.** The training-free construction of $w(s)$ enables efficient guidance, but suffers from abrupt changes after each feedback session. This particularly impacts early-stage skill learning, when both the latent space and policy are underdeveloped. To mitigate this, we employ a smoothing mechanism: $w_e(s) = (1 - \beta_e) \cdot w(s) + \beta_e \cdot 1$, $\beta_e = \max(0, 1 - k_\beta \cdot \frac{e}{T^e})$, where $k_\beta$ is a hyperparameter controlling the decay of $\beta_e$, $e$ is the skill learning epoch index ($e = 1, 2, \ldots, T^e$), and $T^e$ is the total epoch number. During experiments, we use $w_e(s)$ as the guidance signal in epoch $e$. This introduces a smooth transition from pure exploration to guided exploration, with a smaller $k_\beta$

*Table 1.* Safe state coverage results of COLLIE and baselines. For tasks with additional "good" labels, ① refer to *composite safe coverage*, ② refer to *weighted composite safe coverage*. The orange and gray shading represent the best and oracle performances, respectively. COLLIE achieves superior performance across tasks. Table 7 provides safe state ratio results.

| Method | Ant North | Ant Range | Ant Hole | HalfCheetah Right | Quadruped North | Humanoid Hole |
|---|---|---|---|---|---|---|
| **Oracle** | **1381.40** $\pm$ 150.14 | **620.80** $\pm$ 35.52 | **1295.60** $\pm$ 144.86 | **97.80** $\pm$ 4.87 | **112.60** $\pm$ 18.05 | **75.20** $\pm$ 9.39 |
| DIAYN | -4.20 $\pm$ 0.45 | 4.20 $\pm$ 0.45 | 4.20 $\pm$ 0.45 | 0.00 $\pm$ 0.00 | -4.20 $\pm$ 0.84 | 3.60 $\pm$ 0.55 |
| LSD | -1056.80 $\pm$ 515.24 | -916.80 $\pm$ 589.79 | 933.20 $\pm$ 536.19 | -51.00 $\pm$ 11.77 | -0.60 $\pm$ 12.72 | 4.20 $\pm$ 0.45 |
| METRA | -1425.80 $\pm$ 756.14 | -1247.40 $\pm$ 147.97 | 1179.00 $\pm$ 147.26 | -8.40 $\pm$ 4.16 | -200.80 $\pm$ 77.72 | 21.60 $\pm$ 10.81 |
| DDG* | -429.20 $\pm$ 640.30 | -539.20 $\pm$ 477.69 | 559.40 $\pm$ 443.59 | 10.00 $\pm$ 14.85 | -59.60 $\pm$ 18.42 | 7.00 $\pm$ 1.87 |
| DoDont* | 1307.20 $\pm$ 188.33 | -427.60 $\pm$ 224.09 | 1132.20 $\pm$ 171.13 | 82.80 $\pm$ 9.60 | 115.20 $\pm$ 41.60 | 75.80 $\pm$ 14.45 |
| COLLIE | 1333.20 $\pm$ 129.10 | 362.20 $\pm$ 94.55 | 1149.20 $\pm$ 127.05 | 102.20 $\pm$ 4.32 | 128.40 $\pm$ 44.60 | 80.60 $\pm$ 25.01 |

| Method | Ant Range-North ① | Ant Range-North ② | Ant Hole-North ① | Ant Hole-North ② | HalfCheetah Not-Flip | Safety-Gym Hazard |
|---|---|---|---|---|---|---|
| **Oracle** | **501.40** $\pm$ 45.77 | **842.60** $\pm$ 44.34 | **1143.00** $\pm$ 77.27 | **1771.60** $\pm$ 213.72 | **209.60** $\pm$ 14.03 | **-20.80** $\pm$ 7.56 |
| DIAYN | 4.20 $\pm$ 0.45 | 4.20 $\pm$ 0.45 | 4.20 $\pm$ 0.45 | 4.20 $\pm$ 0.45 | 8.80 $\pm$ 14.10 | -34.80 $\pm$ 11.45 |
| LSD | -916.80 $\pm$ 589.79 | -762.40 $\pm$ 508.72 | 925.20 $\pm$ 463.76 | 1244.20 $\pm$ 651.89 | 71.20 $\pm$ 15.61 | -42.80 $\pm$ 12.46 |
| METRA | -1247.40 $\pm$ 147.97 | -1095.00 $\pm$ 249.60 | 1219.00 $\pm$ 133.38 | 1582.00 $\pm$ 171.94 | 187.20 $\pm$ 10.59 | -34.80 $\pm$ 14.80 |
| DDG* | -583.60 $\pm$ 490.33 | -482.20 $\pm$ 432.05 | 512.60 $\pm$ 457.37 | 765.80 $\pm$ 664.16 | 138.20 $\pm$ 14.45 | -26.40 $\pm$ 13.14 |
| DoDont* | -290.20 $\pm$ 241.79 | -10.00 $\pm$ 259.25 | 1135.40 $\pm$ 294.13 | 1566.60 $\pm$ 414.17 | 195.00 $\pm$ 13.55 | -37.60 $\pm$ 13.22 |
| COLLIE | 380.40 $\pm$ 145.54 | 622.00 $\pm$ 199.28 | 1122.00 $\pm$ 190.43 | 1769.80 $\pm$ 218.95 | 215.60 $\pm$ 3.05 | **-16.00** $\pm$ 10.68 |

resulting in a slower transition. At $k_\beta = \infty$, the algorithm remains pure GSD, while $k_\beta = 0$ corresponds to pure USD. By gradually introducing the guidance signal, this approach improves the stability of skill learning, as validated in the ablation studies in Section 4.4.

# 4. Experiment

We conduct extensive experiments to answer the following questions: *Q1*: Can a training-free guidance signal, constructed from minimal human labels, reliably promote diverse behaviors toward safe and meaningful regions? *Q2*: Is COLLIE effective in complex pixel-based settings, where latent state representations must be inferred from raw pixels? *Q3*: What is the individual contribution of each proposed technique in COLLIE?

## 4.1. Setup

**Domains and guidance tasks.** We evaluate COLLIE on five complex robotic locomotion environments: state-based Ant and HalfCheetah from OpenAI Gym (Todorov et al., 2012; Brockman et al., 2016), pixel-based Quadruped and Humanoid from DMControl (Tassa et al., 2018), and state-based Safety Gym (Ray et al., 2019). To assess COLLIE's alignment with human intent, we design tasks with varying guidance types: **(1) Direction**, where the agent moves towards a specific direction (e.g., *North* and *Right*). **(2) Range**, where the agent explores within a range (*Range*). **(3) Hazard avoidance**, where the agent avoids hazardous areas (*Hole* and *Hazard*). **(4) Unsafe behaviour avoid-**

ance, where the agent avoids unsafe actions (*Not-Flip*). **(5) Composite tasks**, which combine multiple guidance types, requiring hazard avoidance while encouraging directional movement (*Range-North* and *Hole-North*). These tasks are illustrated in Fig. 2, with further details in Appendix G.1.

**Feedback collection.** Following prior works (Lee et al., 2021), we use an oracle teacher for systematic evaluation. The teacher provides feedback based on human-defined task rules, aligning with human intent. A segment is labeled as "bad" if it contains any undesirable state, "good" if all states are desirable, and "neutral" otherwise. This conservative labeling reflects human preferences for safety, disfavoring even brief entries into hazardous regions. Appendix G.3 provides further details.

**Baselines and implementation.** We compare COLLIE with three groups of baselines: **(1) USD methods**, including a mutual information-based method, DIAYN (Eysenbach et al., 2019), and two DSD methods, LSD (Park et al., 2022b) and METRA (Park et al., 2024). **(2) GSD method**, specifically, online variant of DoDont (Kim et al., 2024) (DoDont*) and online variant of DDG (Klemsdal et al., 2021) (DDG*). Since DoDont and DDG require training an instruction network with pre-collected expert data, which is unavailable in our setting, we use online-collected data similar to COLLIE. **(3) an Oracle version of COLLIE** (Oracle), which employs a manually designed $w(s)$ in COLLIE to provide ideal guidance signals, serving as the performance upper bound. For constructing the guidance signal, COLLIE uses 40 labeled segments of length $H = 20$ for most tasks.

*Table 2.* Zero-shot downstream task performance on Ant tasks. We report the average and the best performance of the learned skills.

| Method | Hole (avg) | North (avg) | Range (avg) | Hole (best) | North (best) | Range (best) |
|---|---|---|---|---|---|---|
| **Oracle** | **938.23** $_{\pm\,201.76}$ | **1111.08** $_{\pm\,257.30}$ | **-595.66** $_{\pm\,801.53}$ | **1165.42** $_{\pm\,89.36}$ | **1521.60** $_{\pm\,91.41}$ | **717.70** $_{\pm\,28.89}$ |
| DIAYN | 209.85 $_{\pm\,5.78}$ | -2833.11 $_{\pm\,1539.28}$ | **210.09** $_{\pm\,5.86}$ | 243.62 $_{\pm\,17.53}$ | 215.07 $_{\pm\,1.37}$ | 244.73 $_{\pm\,19.49}$ |
| LSD | -34.05 $_{\pm\,777.39}$ | -2017.02 $_{\pm\,1493.83}$ | -894.28 $_{\pm\,455.99}$ | 1112.01 $_{\pm\,386.15}$ | 1054.53 $_{\pm\,467.55}$ | 474.65 $_{\pm\,180.11}$ |
| METRA | 224.07 $_{\pm\,555.11}$ | -1897.47 $_{\pm\,1431.75}$ | -1149.32 $_{\pm\,275.56}$ | 1193.03 $_{\pm\,30.62}$ | 1249.26 $_{\pm\,104.01}$ | 265.29 $_{\pm\,474.71}$ |
| DDG* | 43.78 $_{\pm\,395.34}$ | -2123.80 $_{\pm\,1078.94}$ | -717.89 $_{\pm\,582.27}$ | 923.51 $_{\pm\,321.67}$ | 349.83 $_{\pm\,1523.92}$ | 489.06 $_{\pm\,111.92}$ |
| DoDont* | 333.40 $_{\pm\,633.35}$ | 277.15 $_{\pm\,1519.21}$ | -1045.95 $_{\pm\,374.46}$ | 1155.58 $_{\pm\,48.08}$ | 1397.12 $_{\pm\,45.58}$ | 540.25 $_{\pm\,62.46}$ |
| COLLIE | **650.67** $_{\pm\,637.03}$ | **1054.25** $_{\pm\,376.79}$ | -913.48 $_{\pm\,715.64}$ | **1251.92** $_{\pm\,72.67}$ | **1460.44** $_{\pm\,53.12}$ | **673.55** $_{\pm\,65.86}$ |

Appendix G.3 provides further details.

**Metrics.** We employ three main metrics for evaluation: **(1) Safe state coverage**, which measures the agent's ability to explore the state space while avoiding hazardous regions and behaviors. Following (Kim et al., 2024), this metric assigns a value $+1, -1$ to safe and unsafe states, and computes state coverage by counting the unique $1\times1$ x-y bins (or 1-unit x-axis bins for HalfCheetah tasks) visited by the agent. **(2) Safe state ratio**, which quantifies the proportion of visited safe bins among all visited bins, serves as a normalized safe state coverage. **(3) Downstream task performance**, which evaluates the utility of learned skills in downstream tasks. For all metrics, we report the average and standard deviation across 5 random seeds. Appendix G.3 provides more details.

**4.2. Main Results**

**Hazard area avoidance with sparse feedback.** We first assess COLLIE's ability to avoid static hazardous areas in state-based Ant and HalfCheetah environments. As shown in Fig. 2, COLLIE successfully learns skills constrained to safe regions in Ant North, Range, Hole, and HalfCheetah Right tasks, while unsupervised baselines explore indiscriminately, often visiting hazardous areas. Tables 1 and 7 quantitatively confirm this, with COLLIE achieving near-oracle performance in safe state coverage and safe state ratio, surpassing all baselines on most tasks. These results demonstrate that our training-free guidance signal provides a robust and effective safety constraint. In contrast, DoDont*, which depends on a trained instruction network, performs worse, likely due to the network's instability with limited feedback data.

**Enhanced exploration efficiency with additional "good" labels.** Beyond merely avoiding bad regions, we evaluate COLLIE in tasks that incorporate both positive ("good") and negative ("bad") feedback labels, specifically in Ant Range-North and Hole-North tasks. This allows for assessing COLLIE's ability to not only avoid hazards but also actively promote exploration toward human-preferred regions. To quantify this, we extend the safe state coverage

*Table 3.* Downstream task performance on the HalfCheetah task. We train a hierarchical controller to select low-level frozen skills, and report the average performance.

| Method | Performance |
|---|---|
| **Oracle** | **47.46** $_{\pm\,33.27}$ |
| DIAYN | 10.43 $_{\pm\,4.61}$ |
| LSD | 32.73 $_{\pm\,31.94}$ |
| METRA | 21.58 $_{\pm\,25.53}$ |
| DDG* | 21.57 $_{\pm\,45.27}$ |
| DoDont* | 30.44 $_{\pm\,37.84}$ |
| COLLIE | **45.26** $_{\pm\,15.78}$ |

metric to *composite safe coverage* and *weighted composite safe coverage*, which assign $+1, +1, -1$ and $+2, +1, -1$ to good, neutral, and bad regions, respectively. As shown in Tables 1, COLLIE surpasses almost all baselines in these tasks. Visualizations in Fig. 2 further show dense and uniform skill trajectory coverage within "good" regions, indicating COLLIE's flexibility as a unified framework for encouraging desired behaviors and while avoiding undesirable ones.

**Unsafe behavior avoidance.** Beyond avoiding static hazardous areas, we further assess COLLIE's capability to prevent dynamic unsafe behaviors. In the HalfCheetah Not-Flip task, the agent must learn diverse locomotion skills while avoiding potentially damaging flipping behaviors. As shown in Table 1, COLLIE achieves the highest safe state coverage. This result shows that COLLIE can effectively avoid not just static hazards but also dynamic undesirable behaviors.

**Effectiveness on pixel-based tasks.** To assess COLLIE's ability in complex environments, we evaluate COLLIE on pixel-based Quadruped and Humanoid environments, each with 100 feedback. As shown in Fig. 2, COLLIE successfully avoids hazards in both the Quadruped North and Humanoid Hole tasks, despite the high-dimensional states. Quantitative results in Tables 1 and 7 align with the visualizations. These results show that the COLLIE's training-free guidance mechanism effectively leverages the coherent latent space, generalizing beyond state-based inputs.

*Table 4.* Safe state coverage results of COLLIE under noisy labels with different levels of noise.

| $R_{\text{error}}$ | Ant North | Ant Range |
|---|---|---|
| 0 | $1333.20 _{\pm 129.10}$ | $362.20 _{\pm 94.55}$ |
| 0.5 | $1184.60 _{\pm 124.84}$ | $369.40 _{\pm 51.74}$ |
| 1 | $1084.20 _{\pm 135.81}$ | $360.80 _{\pm 43.53}$ |

*Table 5.* Safe state coverage results of COLLIE using varying number of feedback labels.

| # of labels | Ant North | Ant Hole |
|---|---|---|
| 40 | $1333.20 _{\pm 129.10}$ | $1149.20 _{\pm 127.05}$ |
| 20 | $1035.60 _{\pm 591.51}$ | $829.60 _{\pm 243.73}$ |
| 10 | $801.40 _{\pm 654.11}$ | $766.20 _{\pm 220.39}$ |

*Table 6.* Safe state coverage results of COLLIE using different distances as constraints.

| | Ant North | HalfCheetah Not-Flip |
|---|---|---|
| COLLIE | $1333.20 _{\pm 129.10}$ | $215.60 _{\pm 3.05}$ |
| COLLIE-L2 | $657.60 _{\pm 432.54}$ | $104.20 _{\pm 21.51}$ |

### 4.3. Downstream Task Performance

We evaluate the utility of skills learned by COLLIE on downstream tasks, both in zero-shot settings and after task-specific hierarchical control. We design two types of tasks: an Ant motion task with safety penalties for entering hazardous regions, and a HalfCheetah goal-reaching task that penalizes unsafe behaviors such as flipping, with details provided in Appendix G.2.

In the Ant task, we evaluate all skills in a zero-shot manner. As shown in Table 2, COLLIE achieves the highest average and best performance across all task variants, demonstrating that the learned skills effectively avoid undesirable states while retaining high mobility and task relevance. For the HalfCheetah task, we train a high-level controller to select from the frozen skill set, detailed in Appendix G.3. Table 3 shows that COLLIE outperforms all baselines, confirming that the learned skills are effective for hierarchical downstream task solving. These results demonstrate that COLLIE acquires semantically meaningful and useful skills, enabling strong downstream performance.

### 4.4. Ablation Study

**Evaluation under noisy feedback.** Practical human feedback often contains noise. To demonstrate the robustness of COLLIE to noisy feedback, we evaluate COLLIE under noisy labeling conditions. To simulate human labeling errors, we randomly assign labels (neutral or bad) to states within a band of width $R_{\text{error}}$ around the safety boundaries, reflecting potential human uncertainty in these regions. $R_{\text{error}} = 0$ means that there is no labeling noise. As shown in the Table 4 and Fig. 4, COLLIE remains robust under noisy labels, indicating the reliability of our training-free guidance mechanism.

**Ablation of feedback number $N_{\text{total}}$.** To evaluate the impact of sample sparsity on the performance of COLLIE, we evaluate COLLIE with different numbers of samples. As shown in the Table 5 and Fig. 5, COLLIE effectively aligns with human intent even with only 10 or 20 labels, while its performance improves as the number of labels increases.

**Ablation on the semantically coherent latent space.** To demonstrate the importance of the semantic coherence prop-

erty in COLLIE, we compare COLLIE with its variant (COLLIE-L2), which uses the Euclidean distance between raw states as the distance constraint in the DSD framework, i.e. $d(x, y) = \|x - y\|_2$ in Eq. 1. As shown in the Table 6, using Euclidean distance performs worse than COLLIE, which uses the semantically coherent latent space. This highlights the importance of semantically coherent latent representations.

**Other ablations.** We also conduct ablation studies on query strategy, segment length $H$, and smoothing parameter $k_\beta$, as detailed in Appendix D. These ablations demonstrate the superiority of our query strategy, the robustness to variations in segment length, and the critical role of smoothing.

## 5. Related Work

**Unsupervised skill discovery (USD).** Unsupervised skill discovery (USD) aims to learn a set of distinguishable policies that collectively cover the state space using unlabeled data, without task-specific rewards, to facilitate downstream tasks. A common USD objective is maximizing the mutual information (MI) $I(s, z)$ between states $s$ and latent skills $z$ (Eysenbach et al., 2019; Sharma et al., 2020; Liu & Abbeel, 2021a; Laskin et al., 2022). Though it can yield diverse behaviors, maximizing MI often fails to promote broad exploration, leading to static behaviors (Park et al., 2022b; 2024).

To address this, distance-maximizing skill discovery (DSD) methods are introduced, which establish a connection between latent-space distances and state-space distances to enhance state coverage. METRA (Park et al., 2024) formally derives the DSD objective by replacing MI with the Wasserstein dependency measure. DSD allows any distance function $d(\cdot, \cdot) : \mathcal{S} \times \mathcal{S} \to \mathbb{R}_0^+$ to encourage exploration of different state sub-spaces. Examples include Euclidean distance to encourage geometrically longer travel (Park et al.,

2022b), negative log-likelihoods of an estimated transition probability to prioritize rarely visited states (Park et al., 2023), and temporal distance to encourage temporally distant exploration (Park et al., 2024). However, these methods often lead to uniform exploration within sub-spaces, which may result in unnecessary or even unsafe exploration.

**Guided skill discovery (GSD).** Recent works in GSD incorporate prior knowledge to reduce unnecessary exploration in USD, leveraging expert trajectories (Klemsdal et al., 2021; Kim et al., 2024) or analytical constraint formulas (Kim et al., 2023). Specifically, Klemsdal et al. (2021) and DoDont (Kim et al., 2024) train classifiers to distinguish expert trajectories from others. Klemsdal et al. (2021) further uses the classifier's encoder as a state projection to encourage exploring the expert-concerned state subspace, while DoDont uses the classifier's probability output as the distance function in DSD. Kim et al. (2023) employs Lagrangian Q-learning to ensure skill safety. Moreover, recent studies explore utilizing pairwise human preferences (Hussonnois et al., 2023; 2025) to learn a human-aligned reward model. These models then guide the skill discovery process by identifying preferred regions (Hussonnois et al., 2023) or encouraging alignment between skills and human values (Hussonnois et al., 2025). Despite these advancements, deriving expert trajectories or constraint formulas is often challenging and impractical in complex tasks, and classifiers or reward models trained on limited data can be unstable, especially when expert knowledge is unavailable. This paper aims to address these limitations.

## 6. Conclusion

In this paper, we propose COLLIE, a training-free guided skill discovery framework that effectively aligns exploration with human intent using sparse feedback in scenarios without a high-quality pre-collected dataset. By enforcing a semantically coherent skill latent space with dense unsupervised data, COLLIE constructs a dense guidance signal from minimal human feedback, and integrates this guidance signal into DSD objectives, eliminating the need for expert demonstrations or auxiliary model training. Also, as the semantic coherence property is derived from dense unsupervised data, the latent space is well-structured, ensuring effectiveness even when relying on only sparse human feedback. COLLIE further employs an active query strategy to ensure the guidance signal's accuracy. Experiments show that COLLIE learns diverse, human-preferred skills, avoids unsafe behaviors, and facilitates downstream tasks.

## Acknowledgements

This work is supported by NSFC (No. 62125304), the National Key Research and Development Program of China (2022YFA1004600), the 111 International Collaboration Project (B25027), the Beijing Natural Science Foundation (L233005), and BNRist project (BNR2024TD03003).

## Impact Statement

This paper presents work that aims to advance the field of machine learning, specifically in unsupervised skill discovery. Our proposed method, COLLIE, enables more efficient alignment of AI behaviors with human intentions while reducing the requirement for expert knowledge and avoiding hazardous exploration. By enabling training-free guidance from sparse human labels, our approach lowers barriers for developing robotic systems in complex real-world environments. This could accelerate the deployment of AI systems in industrial domains, especially where safety is important, such as assistive robotics, autonomous vehicles, and industrial automation, ultimately benefiting society through more reliable and trustworthy AI applications.

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

## A. Algorithm

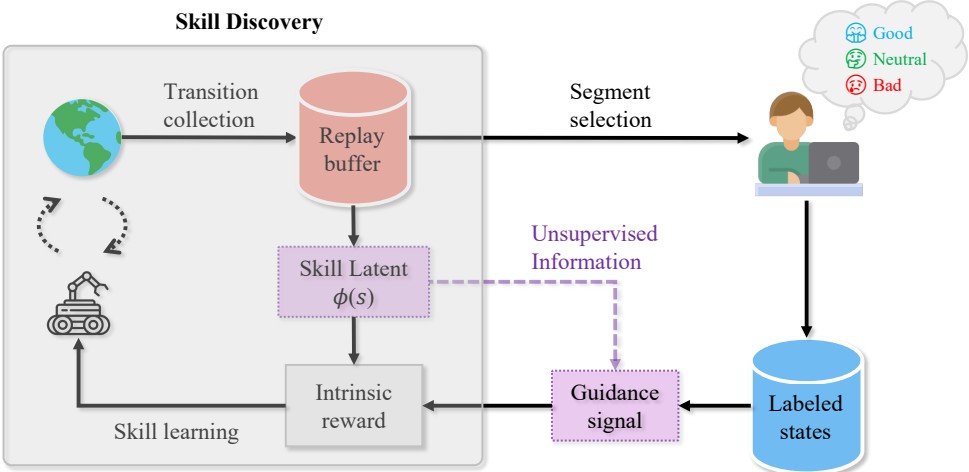

*Figure 3.* An overview of COLLIE. COLLIE performs skill discovery and guidance signal construction simultaneously. The guidance signal is constructed by leveraging both rich unsupervised information from the learned skill latent space and human feedback on labeled states, without relying on expert-level data.

We illustrate the full process of COLLIE in Algorithm 1 and Fig. 3, and illustrate the query selection strategy in detail in Algorithm 2.

---

**Algorithm 2** QUERY SELECTION

---

**Require:** Number of candidate queries $N_c$, number of queries per feedback session $M$, feedback buffer $\mathcal{D}$
 1: **if** Feedback buffer $\mathcal{D}$ is empty **then**
 2:      Sample $M$ segments $\{\sigma_i\}_{i=1}^{N_c}$ and return
 3: **else**
 4:      Sample $N_c$ segments $\{\sigma_i\}_{i=1}^{N_c}$
 5:      Initialize query selection vector of shape $N_c$ with zeros: $\hat{I} = [0, 0, \ldots, 0]$.
 6:      **for** each segment $\sigma_i$ **do**
 7:          Calculate selection score and store it in $\hat{I}$: $\hat{I}_i \leftarrow I(\sigma_i)$
 8:      **end for**
 9:      Select $M$ queries with the top-$M$ query selection score $\hat{I}$
10: **end if**

---

This online update, shown in Algorithm 1 and Fig. 3, differentiates our method from existing GSD approaches by using guidance derived from expert trajectories or analytical constraints. The online update allows the guidance signal to evolve jointly with the skills, thereby leveraging the unsupervised skill latent space and the online collected trajectories. Specifically, the unsupervised skill latent space is well-structured, which enables training-free construction of the guidance signal and ensures reliability with sparse human feedback. Additionally, trajectories collected alongside the skill discovery enable interactive query collection, eliminating the need for expert trajectories to train the guidance model, which may fail when the selected trajectories are of low quality.

# B. Proof

In this section, we theoretically analyze the performance of the proposed guidance signal $w(s)$. Since $w(s)$ is derived as an expectation in Eq. 7, we evaluate it by analyzing the estimated human desirability distribution, given as:

$$\text{softmax}\left([-d_\phi(s, \mathcal{D}_0), \ -d_\phi(s, \mathcal{D}_1), \ -d_\phi(s, \mathcal{D}_2)]\right). \tag{15}$$

Denote the labeled dataset as $\mathcal{D} = \{(s_i, y_i)\}_{i=1}^n$, where $y_i \in \{0, 1, 2\}$ represents the human desirability label. For each class $k$, let $\mathcal{D}_k = \{(s_i, y_i) \in \mathcal{D} \mid y_i = k\}$. Let the true human desirability function $g(s) : \mathcal{S} \to \{0, 1, 2\}$ be a random variable, with conditional probabilities $\eta_k(s) = P(g(s) = k \mid s)$ for $k \in \{0, 1, 2\}$. We consider a classifier $\hat{g}(s)$ induced from $w(s)$. For any given $s$, $\hat{g}(s)$ calculates the distances $d_{\phi,k}(s) = \min_{(s_i, y_i) \in \mathcal{D}_k} \|\phi(s_i) - \phi(s)\|$, estimates the probabilities as $\hat{\eta}_k(s) = \frac{\exp(-d_{\phi,k}(s))}{\sum_{j=0}^2 \exp(-d_{\phi,j}(s))}$, and assigns the state $s$ to the class $\hat{g}(s) = \arg\max_k \hat{\eta}_k(s)$. The Bayes error rate of this classification problem is defined as $P^*(s) = 1 - \max_k \eta_k(s)$.

In addition, we make the following assumptions:

- The labeled dataset $\mathcal{D} = \{(s_i, y_i)\}_{i=1}^n$ is sampled independently and identically distribute.

- The number of samples in each class, $n_k = |\mathcal{D}_k|$, satisfies $n_k \to \infty$ as $n \to \infty$.

**Proposition 3.1.** *Consider the induced classifier $\hat{g}(s)$ derived from the guidance signal $w(s)$, i.e., $\hat{g}(s) = \arg\max_k \frac{\exp(-d_\phi(s, \mathcal{D}_k))}{\sum_{j=0}^2 \exp(-d_\phi(s, \mathcal{D}_j))}$. The asymptotic expected error rate of the classifier $\hat{g}(s)$ is bounded by the Bayes error rate $P^*(s)$, which is formally expressed as:*

$$P(\hat{g}(s) \neq g(s)) \leq 2P^*(s) - \frac{3}{2}[P^*(s)]^2. \tag{8}$$

*Proof.* Consider the nearest neighbor classifier $\hat{g}_{\text{NN}}(s)$, which selects the label $y^*$ of the labeled point $s^*$ closest to $s$:

$$s^* = \arg\min_{s_i \in \mathcal{D}} \|\phi(s_i) - \phi(s)\|, \quad \hat{g}_{\text{NN}}(s) = y^*. \tag{16}$$

For any $(s^*, y^*)$ selected by $\hat{g}_{\text{NN}}(s)$, we have $d_{\phi,y^*}(s) \leq d_{\phi,i}(s), \forall (s_i, y_i) \in \mathcal{D}$. Then, $\hat{\eta}_{y^*}(s) \geq \hat{\eta}_k(s), \forall k \in \{0, 1, 2\}$. Therefore, $\hat{g}(s)$ always has the same estimation as the nearest neighbor classifier $\hat{g}_{\text{NN}}(s)$. The classifier $\hat{g}(s)$ is equivalent to a nearest neighbor classifier $\hat{g}_{\text{NN}}(s)$.

Next, we study the asymptotic expected error rate of the nearest neighbor classifier $\hat{g}_{\text{NN}}(s)$ for this 3-class classification problem. The analysis is inspired by Cover & Hart (1967).

In this 3-class classification problem, the classification error occurs when $y^* \neq g(s)$. Given $s$ and its nearest neighbor $s^*$, the error probability is:

$$P(y^* \neq g(s) \mid s, s^*) = 1 - P(y^* = g(s) \mid s, s^*). \tag{17}$$

Note that conditional on $s$, the distribution of $g(s)$ is determined by $\eta_k(s)$. Since $y^*$ is a realization of $g(s^*)$, conditional on $s^*$, the distribution of $y^*$ is determined by $\eta_k(s^*)$. As $y^*$ and $g(s)$ are conditionally independent given $s$ and $s^*$, we have

$$P(y^* = g(s) \mid s, s^*) = \sum_{k=0}^2 P(g(s) = k \mid s)P(y^* = k \mid s^*) = \sum_{k=0}^2 \eta_k(s)\eta_k(s^*), \tag{18}$$

Taking expectation over $s$ and $s^*$, we have

$$P(\hat{g}_{\text{NN}}(s) \neq g(s)) = \mathbb{E}\left[1 - \sum_{k=0}^2 \eta_k(s)\eta_k(s^*)\right]. \tag{19}$$

As $n \to \infty$, $s^*$ converges to $s$ (due to denseness of the point set), and if $\eta_k$ is continuous, then $\eta_k(s^*) \to \eta_k(s)$. Therefore, the asymptotic error rate becomes

$$P(\hat{g}_{\text{NN}}(s) \neq g(s)) = \mathbb{E}\left[1 - \sum_{k=0}^2 \eta_k(s)^2\right]. \tag{20}$$

To bound this expression, let $\eta_{(0)} \geq \eta_{(1)} \geq \eta_{(2)}$ be the ordered values of $\eta_k(s)$, so $\max_k \eta_k(s) = \eta_{(0)}$ and $P^*(s) = 1 - \eta_{(0)}$. The sum of squares $\sum \eta_k(s)^2$ is minimized when the remaining probability $1 - \eta_{(0)}$ is distributed uniformly among the other 2 classes. Therefore, we have

$$\sum_{k=0}^{2} \eta_k(s)^2 \geq \eta_{(0)}^2 + \frac{(1 - \eta_{(0)})^2}{2}. \tag{21}$$

Using $P^*(s) = 1 - \eta_{(0)}$, we have

$$1 - \sum_{k=0}^{2} \eta_k(s)^2 \leq 2P^*(s) - \frac{3}{2}[P^*(s)]^2. \tag{22}$$

Substituting it into Eq. 20, we derive that for this 3-class classification problem, as $n \to \infty$, the error rate of the nearest neighbor classifier satisfies:

$$P(\hat{g}_{\text{NN}}(s) \neq g(s)) \leq 2P^*(s) - \frac{3}{2}[P^*(s)]^2. \tag{23}$$

Therefore, $\hat{g}(s)$ satisfies

$$P(\hat{g}(s) \neq g(s)) \leq 2P^*(s) - \frac{3}{2}[P^*(s)]^2, \tag{24}$$

which concludes the proof. $\square$

## C. Derivation of the Objective Function

In this section, we derive the objective function of COLLIE, Eq. 10, from the DSD-form objective function, Eq. 9, in a similar manner as Kim et al. (2024). We assume the guidance signal $w(s)$ is continuous.

We start by restating Eq. 9:

$$\sup_{\pi,\phi} \mathbb{E}_{\tau,z}\left[\sum_{t=0}^{T-1}(\phi(s_{t+1}) - \phi(s_t))^\top z\right] \text{ s.t. } \|\phi(s') - \phi(s)\|_2 \leq w(s)d(s,s'), \ \forall(s,s') \in S_{\text{adj}}. \tag{25}$$

Define a scaled latent function $\phi'(s) \triangleq \frac{\phi(s)}{w(s)}$. As $w(s) > 0$ in COLLIE, we derive the following formula approximately.

$$\sup_{\pi,\phi} \mathbb{E}_{\tau,z}\left[\sum_{t=0}^{T-1}(\phi(s_{t+1}) - \phi(s_t))^\top z\right] \text{ s.t. } \|\phi'(s') - \phi'(s)\|_2 \leq d(s,s'), \ \forall(s,s') \in S_{\text{adj}}. \tag{26}$$

The approximation $\frac{\phi(s')}{w(s)} \approx \frac{\phi(s')}{w(s')}$ leverages the continuity of the guidance signal $w(s)$, as $s$ and $s'$ are adjacent states in this context.

Then, we replace $\phi(s)$ with $w(s)\phi'(s)$, deriving

$$\sup_{\pi,\phi} \mathbb{E}_{\tau,z}\left[\sum_{t=0}^{T-1} w(s_t)(\phi'(s_{t+1}) - \phi'(s_t))^\top z\right] \text{ s.t. } \|\phi'(s') - \phi'(s)\|_2 \leq d(s,s'), \ \forall(s,s') \in S_{\text{adj}}. \tag{27}$$

which is exactly Eq. 10.

## D. More Experimental Results

**Safe state ratio results for Section 4.2.**   We report the safe state ratio results in Table 7. For tasks with additional good labels (Ant Range-North and Ant Hole-North), the safe ratio is calculated as the proportion of good and neutral state bins relative to all visited state bins. As shown in the table, COLLIE outperforms other baselines in 7 out of 10 tasks. Note that DIAYN achieves 100% safe state ratio in Ant Hole and Ant Hole-North tasks. This is primarily because the range of states covered by DIAYN is highly restricted, causing it to visit only good or neutral states. In contrast, COLLIE achieves a superior safe coverage (in Table 1) and safe state ratio simultaneously, demonstrating its effectiveness.

*Table 7.* Safe state ratio results (%) of COLLIE and baselines. The orange and gray shading represent the best and oracle performances, respectively. COLLIE achieves superior performance across tasks.

| Method | Ant North | Ant Range | Ant Hole | HalfCheetah Right | Quadruped North | Humanoid Hole |
|---|---|---|---|---|---|---|
| **Oracle** | **92.60** ± 1.60 | **94.30** ± 2.00 | **98.90** ± 0.70 | **99.00** ± 0.00 | **79.80** ± 2.90 | **91.20** ± 5.30 |
| DIAYN | 0.00 ± 0.00 | 0.00 ± 0.00 | **100.00** ± 0.00 | 50.00 ± 0.00 | 6.20 ± 8.50 | **100.00** ± 0.00 |
| LSD | 18.30 ± 10.90 | 36.40 ± 26.10 | 69.00 ± 13.50 | 28.50 ± 4.20 | 20.60 ± 24.20 | **100.00** ± 0.00 |
| METRA | 20.40 ± 14.70 | 23.90 ± 2.10 | 74.70 ± 2.20 | 48.00 ± 1.00 | 21.10 ± 10.60 | 82.30 ± 15.40 |
| DDG* | 20.10 ± 16.80 | 28.10 ± 8.80 | 73.90 ± 9.40 | 52.70 ± 3.30 | 11.30 ± 14.30 | **100.00** ± 0.00 |
| DoDont* | 93.40 ± 4.70 | 36.70 ± 5.70 | 83.20 ± 4.50 | 86.70 ± 5.30 | 76.00 ± 8.00 | 84.20 ± 4.30 |
| COLLIE | **96.90** ± 2.50 | **80.90** ± 7.80 | 90.60 ± 2.40 | **98.70** ± 0.50 | **87.60** ± 7.50 | 90.50 ± 9.80 |

| Method | Ant Range-North | Ant Hole-North | HalfCheetah Not-Flip | Safety-Gym Hazard | | |
|---|---|---|---|---|---|---|
| **Oracle** | **95.80** ± 1.50 | **98.60** ± 0.30 | **100.00** ± 0.00 | **37.00** ± 4.70 | | |
| DIAYN | 0.00 ± 0.00 | **100.00** ± 0.00 | 75.20 ± 14.00 | 28.30 ± 7.20 | | |
| LSD | 36.40 ± 26.10 | 74.50 ± 1.50 | 76.10 ± 5.80 | 23.20 ± 7.80 | | |
| METRA | 23.90 ± 2.10 | 75.60 ± 2.50 | 89.20 ± 3.20 | 28.30 ± 9.30 | | |
| DDG* | 40.20 ± 33.50 | 76.90 ± 16.40 | **100.00** ± 0.00 | 33.50 ± 8.20 | | |
| DoDont* | 41.30 ± 6.20 | 83.90 ± 5.00 | 91.70 ± 2.80 | 26.50 ± 8.30 | | |
| COLLIE | **81.40** ± 8.50 | 93.10 ± 4.80 | **100.00** ± 0.00 | **40.00** ± 6.70 | | |

**Evaluation under noisy feedback.**   Practical human feedback often contains noise. To demonstrate the robustness of COLLIE to noisy feedback, we evaluate COLLIE under noisy labeling conditions. To simulate human labeling errors, we randomly assign labels (neutral or bad) to states within a band of width $R_{error}$ around the safety boundaries, reflecting potential human uncertainty in these regions. $R_{error} = 0$ means that there is no labeling noise. As shown in the Table 4 and Fig. 4, COLLIE remains robust under noisy labels, indicating the reliability of our training-free guidance mechanism.

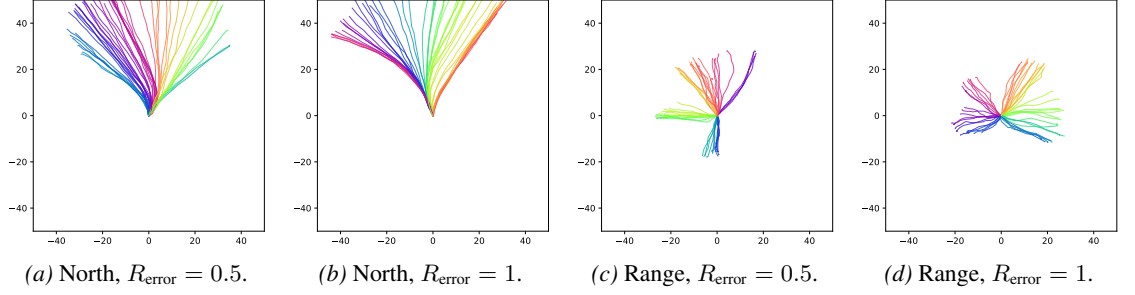

| (a) North, $R_{error} = 0.5$. | (b) North, $R_{error} = 1$. | (c) Range, $R_{error} = 0.5$. | (d) Range, $R_{error} = 1$. |

*Figure 4.* Visualizations of skills learned by COLLIE under noisy feedback with varying $R_{error}$ on Ant North and Range tasks.

**Ablation of feedback number $N_{total}$.**   To evaluate the impact of sample sparsity on the performance of COLLIE, we evaluate COLLIE with different numbers of samples. As shown in the Table 5 and Fig. 5, COLLIE effectively aligns with human intent even with only 10 or 20 labels, while its performance improves as the number of labels increases.

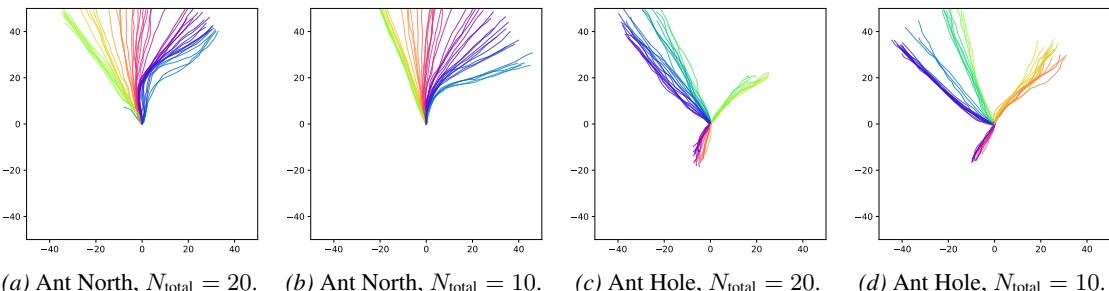

*(a)* Ant North, $N_{\text{total}} = 20$.    *(b)* Ant North, $N_{\text{total}} = 10$.    *(c)* Ant Hole, $N_{\text{total}} = 20$.    *(d)* Ant Hole, $N_{\text{total}} = 10$.

*Figure 5.* Visualizations of skills learned by COLLIE with varying feedback number $N_{\text{total}}$.

**Ablation on query selection methods.** To evaluate the effectiveness of COLLIE's query selection method, we compare it with three baselines: a uniform sampling method (Uniform), an uncertainty-focused method (Uncertainty) that prioritizes states with large guidance signal entropy:

$$H_{\text{w}}(s) = \mathbb{H}\big[\text{softmax}\big([-d_\phi(s, \mathcal{D}_i)]_{i=0}^2\big)\big], \tag{28}$$

and a variant of COLLIE (denoted as COLLIE-$\phi$) that maximizes the entropy of labeled states in the latent space:

$$H_{\text{latent}}(s) = -\log \Pr_{s \sim D}(\phi(s)). \tag{29}$$

As shown in Table 8, COLLIE consistently outperforms all three variants, achieving higher safe state coverage across all tasks. While the Uncertainty approach refines preference boundaries, it often neglects exploration and fails to cover diverse regions of the state space. In contrast, COLLIE prioritizes under-explored regions, ensuring effective guidance across a wider state space under sparse feedback. These results underscore the critical role of exploration in query selection of GSD. For the COLLIE-$\phi$ method, it aims to leverage the well-structured latent space to enhance the query selection method, which improves the accuracy of $w(s)$ by selecting states that can best cover the latent space. However, its effectiveness is limited in some complex tasks, such as Ant Hole, perhaps because the latent space is not yet fully developed in such complex state spaces during early training stages. Therefore, for the robustness of the proposed method, we use the state space entropy instead of skill latent space entropy in COLLIE.

*Table 8.* Comparison of safe state coverage of COLLIE and various query selection methods.

| Method | Ant Hole | Ant North | Ant Range |
|---|---|---|---|
| COLLIE | $1149.20_{\pm 127.05}$ | $1333.20_{\pm 129.10}$ | $329.80_{\pm 110.41}$ |
| Uniform | $633.80_{\pm 279.41}$ | $1257.80_{\pm 189.01}$ | $40.80_{\pm 130.20}$ |
| Uncertainty | $1034.40_{\pm 468.49}$ | $1059.20_{\pm 79.16}$ | $-79.60_{\pm 302.86}$ |
| COLLIE-$\phi$ | $951.40_{\pm 237.30}$ | $1285.40_{\pm 256.10}$ | $344.40_{\pm 53.39}$ |

**Ablation of the smooth parameter.** To evaluate the impact of the smoothing parameter $k_\beta$, which controls the transition speed from USD to GSD, we conduct an ablation study. Table 9 shows that incorporating $k_\beta$ improves performance, but overly slow transitions weaken the guidance signals, impairing skill learning. Based on these results, we set $k_\beta = 5$ in the main experiments, as it consistently outperforms the configuration without smoothing.

*Table 9.* Safe state coverage results of COLLIE using different smoothing parameter $k_\beta$.

| $k_\beta$ | Ant Hole | Ant North | Ant Range |
|---|---|---|---|
| 3 | $1041.60_{\pm 226.01}$ | $1349.00_{\pm 162.01}$ | $312.40_{\pm 59.77}$ |
| 5 | $1149.20_{\pm 127.05}$ | $1333.20_{\pm 129.10}$ | $362.20_{\pm 94.55}$ |
| 10 | $1068.40_{\pm 321.47}$ | $1251.80_{\pm 99.38}$ | $399.40_{\pm 65.12}$ |
| $\infty$ | $991.60_{\pm 188.15}$ | $1120.80_{\pm 420.01}$ | $320.00_{\pm 91.07}$ |

**Ablation of segment length $H$.** We evaluate COLLIE with varying segment lengths $H$. As shown in the Table 10 and Fig. 6, COLLIE consistently achieves superior performance across different segment lengths ($H = 20, 40, 60$), demonstrating its robustness to this parameter.

*Table 10.* Safe state coverage results of COLLIE with different segment length $H$.

| $H$ | Ant North | Ant Range |
|----|-----------|-----------|
| 20 | $1333.20 _{\pm 129.10}$ | $362.20 _{\pm 94.55}$ |
| 40 | $1325.20 _{\pm 75.70}$ | $309.25 _{\pm 77.00}$ |
| 60 | $1327.60 _{\pm 132.60}$ | $347.00 _{\pm 35.24}$ |

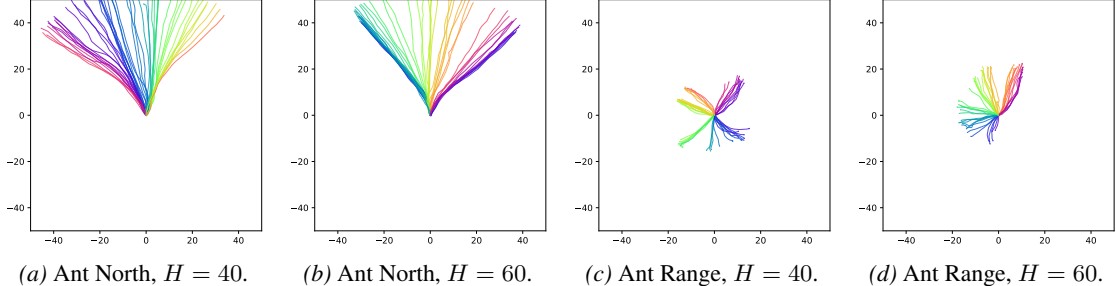

*(a)* Ant North, $H = 40$.    *(b)* Ant North, $H = 60$.    *(c)* Ant Range, $H = 40$.    *(d)* Ant Range, $H = 60$.

*Figure 6.* Visualizations of skills learned by COLLIE with varying segment length $H$.

**Evaluation under noisy observation.** To evaluate COLLIE's robustness to state observation noise, we add Gaussian noise to the observed state: for an original state $s$, the observed input is $\hat{s} = (1 + x) \cdot s$, where $x \sim \mathcal{N}(0, 0.01I)$, and $\mathcal{N}$ denotes the Gaussian distribution. As shown in the table below, the performance degrades only slightly, demonstrating that COLLIE remains effective under moderate observation noise.

*Table 11.* Safe state coverage results of COLLIE under noisy observation.

| Method | Ant Range |
|--------|-----------|
| COLLIE | $362.20 _{\pm 94.55}$ |
| COLLIE (w/ noise) | $320.80 _{\pm 83.15}$ |

**Ablation of additive reward function.** COLLIE employs a multiplicative intrinsic reward function, which is derived by integrating the guidance signal as a distance constraint within the DSD framework. To assess the efficacy of this reward form, we conducted an ablation study using the additive variant: $r = (\phi(s') - \phi(s))^T z + w(s)$. As shown in the table below, the additive reward degrades performance, confirming that our multiplicative formulation better preserves the theoretical grounding established in Sections 3.2-3.3.

*Table 12.* Safe state coverage results of COLLIE with different reward form.

| Method | Ant North | Ant Hole |
|--------|-----------|----------|
| COLLIE | $1333.20 _{\pm 129.10}$ | $1149.20 _{\pm 127.05}$ |
| COLLIE (additive reward) | $1160.80 _{\pm 157.70}$ | $354.40 _{\pm 177.82}$ |

**Human experiments.** We engage human labelers to provide feedback on visualized 2D trajectories in the Ant-Range task, who are instructed by task descriptions in Appendix G.1. We conduct five runs with different seeds, collecting 40 human labels per run. As shown in the Table 13 and Fig. 7, COLLIE consistently achieved high performance, confirming its practical effectiveness with real human input.

*Table 13.* Safe state coverage results of COLLIE with the oracle teacher in Section 4.1 and with real human labelers.

| Method | Ant Range |
| --- | --- |
| COLLIE (oracle teacher) | $362.20_{\pm 94.55}$ |
| COLLIE (human labelers) | $361.40_{\pm 49.95}$ |

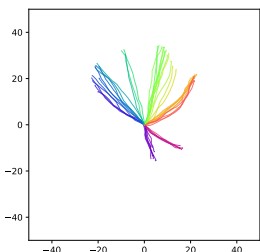

*Figure 7.* Visualizations of skills learned by COLLIE with real human labelers on the Ant Range task.

## E. Discussion and Limitation

**Dead zone phenomenon in scenarios with obstacles.** Despite COLLIE's strong performance, we identified under-explored regions in complex scenarios like "Hole". As shown in Fig. 2, both COLLIE and DoDont* fail to explore areas behind the holes in the Ant Hole task. Even with accurate guidance, as demonstrated by the "Oracle" results, the base skills fail to bypass obstacles and reach areas hidden behind them.

We believe this is primarily due to the inherent exploration mechanisms of the underlying METRA framework. In scenarios with obstacles, METRA's optimal behaviors do not encourage skills to explore regions behind obstacles, resulting in "dead zones".

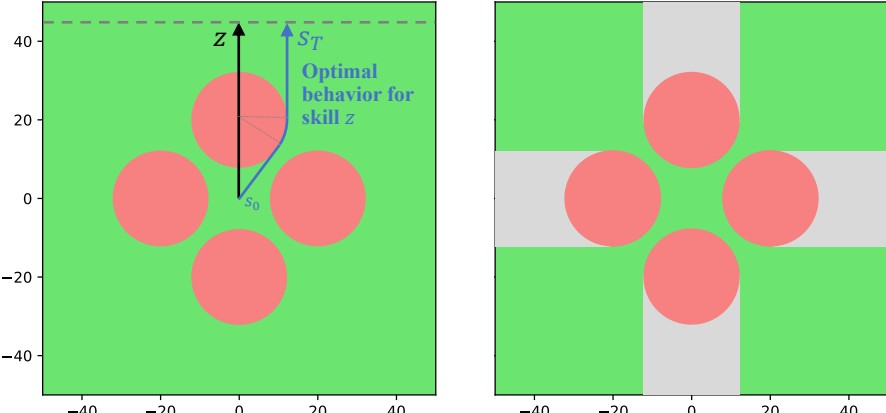

*Figure 8.* Visualization of the reason for the dead zone phenomenon in scenarios with obstacles (Hole as an example). Human-undesirable regions are highlighted in red, and other regions are highlighted in green. In the right subfigure, gray regions represent "dead zones".

We use Fig. 8 to illustrate this phenomenon. Consider a mass point in a "Hole" scenario, with a 2-dimensional skill space, skills are expected to move in diverse directions. If we ignore the human-undesirable regions, the METRA's latent space $\phi(s)$ aligns with the 2D state space, i.e., $\phi(s) = s$. Since COLLIE's objective (Eq. 10) differs from METRA only by reweighting, if the samples to train the latent space are sufficient and the non-human-undesirable regions are well explored, the latent space COLLIE learned will be equivalent to METRA learned in the non-human-undesirable regions.

Consequently, for the specific skill latent $z$ shown in Fig. 8, the optimal behavior follows the blue trajectory, which achieves the largest reward $(\phi(s') - \phi(s))^T z$ within the fixed timesteps. The trajectory of such optimal behavior will go parallel with the skill latent $z$ after passing the human-undesirable regions, which makes the region just behind the hole a dead zone.

The main reason for the dead zone is that METRA uses the DSD framework only to encourage the coverage of skills in the state space by aligning the trajectory with the uniformly distributed skill latent $z$, without using any pure exploration schemes such as prediction errors (Pathak et al., 2017; Burda et al., 2019), state entropy (Hazan et al., 2019; Liu & Abbeel, 2021b), or pseudo-counts (Bellemare et al., 2016; Ostrovski et al., 2017). This makes METRA easily ignore the under-explored areas, even if they are close to the learned skills and are not hard to reach. Combining the DSD framework with exploration schemes is a promising aspect for addressing the dead zone phenomenon.

**Evaluation on increasing skill latent dimension.** To investigate whether under-explored regions result from the limited representational capacity of the skill space, we increased the skill latent dimension from 2 to 4 to enhance its ability to encode diverse behaviors. However, as visualized in Fig. 9, the expanded skill space did not lead to exploration of the occluded regions. This result is consistent with the previous analysis, confirming that the occurrence of under-explored regions is not due to insufficient expressive capacity of the skill space, but stems from inherent limitations of the METRA algorithm.

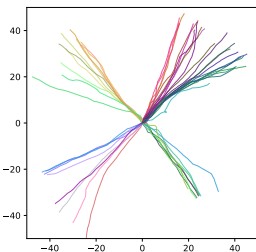

*Figure 9.* Visualizations of skills learned by COLLIE using 4-dim skills.

## F. Extended Related Work

**Unsupervised reinforcement learning.** Unsupervised reinforcement learning (Xie et al., 2022) learns a policy or set of policies with unlabeled data (transitions without task-specific rewards) to explore the state space. The target is to acquire knowledge of the environment, thereby facilitating downstream tasks. To encourage exploration, various intrinsic rewards are proposed, such as prediction errors (Pathak et al., 2017; Burda et al., 2019), state entropy (Hazan et al., 2019; Liu & Abbeel, 2021b), pseudo-counts (Bellemare et al., 2016; Ostrovski et al., 2017), and empowerment measures (Eysenbach et al., 2019; Sharma et al., 2020).

**Unsupervised skill discovery.** Skill discovery methods construct intrinsic reward with empowerment measures and learn a set of distinguishable policies to cover the state space jointly. A typical choice for the empowerment measure is the mutual information $I(s, z)$ between the state $s$ and the skill latent $z$. Recent studies explore different mutual information formulations. The reverse form $I(s, z) = H(z) - H(z|s)$ (Eysenbach et al., 2019; Park et al., 2022b) trains an additional skill discriminator $q(z|s)$ to encourage skills to visit different states. The forward form $I(s, z) = H(s) - H(s|z)$ (Sharma et al., 2020; Liu & Abbeel, 2021a; Laskin et al., 2022) trains an additional state density model $q(s|z)$ for each skill, enabling integration with model-based RL algorithms. Though maximizing the mutual information could induce diverse behaviors, this does not encourage exploration, which may lead to static behaviors (Park et al., 2022b; 2024).

To address this issue, recent studies have explored various strategies, such as employing exploration methods to collect diverse trajectories (Campos et al., 2020), incorporating an entropy maximization term into the intrinsic reward (Liu & Abbeel, 2021a), and eliminating the anti-exploration term from mutual information in skill learning (Zheng et al., 2025). An outstanding category is the distance-maximizing skill discovery approach (DSD) (Park et al., 2023), which links the distance in latent space with that in space to encourage coverage in state space. The objective function of DSD is formally derived in METRA (Park et al., 2024) by replacing the traditional mutual information objective in SD with the Wasserstein dependency measure (WDM). The distance could be any arbitrary function $d(\cdot, \cdot) : \mathcal{S} \times \mathcal{S} \to \mathbb{R}_0^+$ to encourage exploring state sub-space with different properties. For example, Euclidean distance (Park et al., 2022b) encourages geometrically longer travel, the negative log-likelihood of an estimated transition probability (Park et al., 2023) encourages visiting rarely visited states, and temporal distance (Park et al., 2024) encourages temporally distant exploration. However, these DSD methods do not consider human desirability when exploring, which makes the exploration inefficient when the state space is vast and complex.

**Guided skill discovery.** Recent studies mitigate unnecessary exploration in unsupervised skill discovery by incorporating prior knowledge into skill learning. The prior knowledge could come from expert trajectories (Klemsdal et al., 2021; Kim et al., 2024) and analytical formulas of constraints (Kim et al., 2023). Specifically, Klemsdal et al. (2021) and DoDont (Kim et al., 2024) train a classifier to distinguish expert trajectories from other trajectories. Klemsdal et al. (2021) further uses the encoder of the classifier as a state projection to encourage exploring the expert-concerned state subspace. While DoDont use the probability output by the classifier as the distance function in DSD. Kim et al. (2023) considers Lagrangian Q learning in skill learning to ensure the safety of learned skills. Recent studies explore utilizing pairwise human preferences (Hussonnois et al., 2023; 2025) to learn a human-aligned reward model. These models then guide the skill discovery process by identifying preferred regions (Hussonnois et al., 2023) or encouraging alignment between skills and human values (Hussonnois et al., 2025). Despite these advancements, deriving expert trajectories or constraint formulas is often challenging and impractical in complex tasks, and classifiers or reward models trained on limited data can be unstable. This paper aims to address these limitations.

**Human feedback in policy learning.** Prior works have demonstrated the effectiveness of integrating human feedback into policy learning to overcome the challenges of manual reward design. Early works such as Daniel et al. (2014) used active queries with numerical ratings, while Akrour et al. (2014) and Sugiyama et al. (2012) leveraged pairwise preferences to iteratively refine policies or reward functions. Wang et al. (2016) explored interactive learning mechanisms in language games through implicit selection feedback. Building on these foundations, preference-based reinforcement learning (PbRL) (Christiano et al., 2017; Mu et al., 2025b) has emerged as a key framework for aligning agents with human intent through structured comparisons. Inspired by these efforts, COLLIE incorporates human guidance to direct exploration toward desirable behaviors.

However, a core limitation of PbRL is the high cost of human supervision. To address this issue, recent PbRL studies have focused on improving feedback efficiency via enhanced query selection (Lee et al., 2021; Shin et al., 2023; Luan et al., 2025), unsupervised pretraining (Lee et al., 2021; Cheng et al., 2024), and data augmentation (Park et al., 2022a; Choi et al., 2024). Despite these efforts, recent studies show that pairwise comparisons, a common approach in PbRL, suffer from segment indistinguishability (Mu et al., 2025c;a), which significantly undermines their effectiveness. Since feedback is sparse in our work, and the pretraining phase involves numerous potential tasks, the issue of indistinguishability is further exacerbated. COLLIE adopts discrete ratings for single segments, similar to Akrour et al. (2014) and Sugiyama et al. (2012), to avoid the indistinguishability problem inherent in pairwise comparisons. Additionally, unlike standard approaches that require training parameterized reward models, COLLIE proposes a training-free method that efficiently utilizes sparse feedback by leveraging the semantic coherence of the unsupervised skill latent space.

# G. Experimental Details

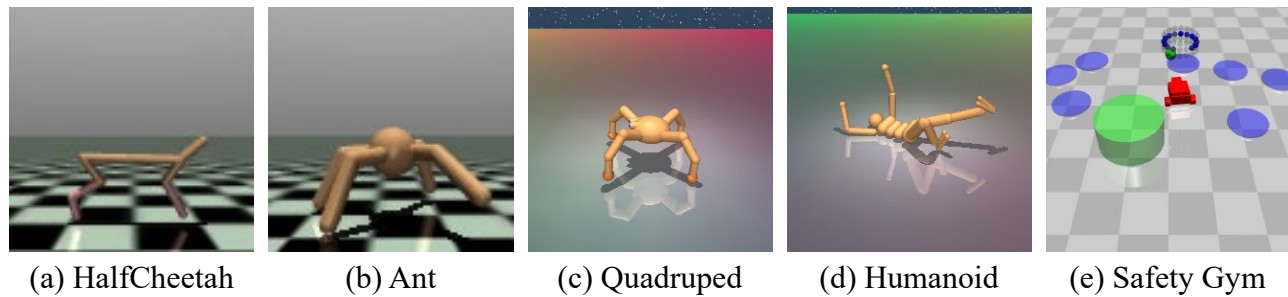

(a) HalfCheetah     (b) Ant     (c) Quadruped     (d) Humanoid     (e) Safety Gym

*Figure 10.* Benchmark environments.

## G.1. Setup

**Environments.** We evaluate COLLIE on five complex robotic locomotion environments: state-based Ant and HalfCheetah from OpenAI Gym (Todorov et al., 2012; Brockman et al., 2016), pixel-based Quadruped and Humanoid from DMControl (Tassa et al., 2018), and state-based Safety Gym (Ray et al., 2019), as illustrated in Fig. 10. In pixel-based DMControl environments, we follow prior works (Park et al., 2022b; 2024; Kim et al., 2024) by using colored floors, allowing the agent to infer its location from pixel observations. The observation is $64 \times 64$ RGB images of the scene for these pixel-based environments. For Safety Gym (Ray et al., 2019), we use a customized `Safexp-CarGoal1-v0` environment, where a car must navigate to a goal while avoiding hazards. To ensure consistency within a single experiment, the locations of hazards are randomly generated at the start of each experiment but remain fixed throughout its duration.

**Guidance task designs.** To assess COLLIE's alignment with human intent, we design tasks with varying guidance types:

- **Direction guidance**, where the agent moves towards a specific direction (*North* and *Right*).

- **Range guidance**, where the agent explores within a range (*Range*).

- **Hazard avoidance**, where the agent avoids hazardous areas (*Hole* and *Hazard*).

- **Unsafe behaviour avoidance**, where the agent avoids unsafe actions (*Not-Flip*).

- **Composite tasks**, which combine multiple guidance types, requiring hazard avoidance while encouraging directional movement (*Range-North* and *Hole-North*).

Tasks are illustrated in Fig. 2. The oracle guidance signals for states are specified as follows:

- **North** (for Ant and Quadruped environments): A state is considered bad if the location $(x, y)$ does not satisfy $y \geq |x|$.

- **Right** (for HalfCheetah environment): A state is considered bad if the location $x$ does not satisfy $x \geq 0$.

- **Range** (for Ant environment): There is a safe area defined as a circle with its center at $(0, 5)$ and radius $r = 20$. A state is considered bad if the agent is outside this safe area.

- **Hole** (for Ant and Humanoid environment): There are four holes in the scene. A state is considered bad if the agent is located in any of these holes: For Ant environment, the holes are circles with centers at $(0, 20)$, $(0, -20)$, $(20, 0)$, $(-20, 0)$, all with a common radius $r = 12$. For Humanoid environment, the holes are circles with centers at $(0, 8)$, $(0, -8)$, $(8, 0)$, $(-8, 0)$, all with a common radius $r = 4$.

- **Hazard** (for Safety-Gym environment): A state is considered bad if the agent is in either hazard area. The locations of hazards are randomly generated at the start of each experiment but remain fixed throughout its duration.

- **Not-Flip** (for HalfCheetah environment): A state is considered bad if the agent flips. Specifically, the agent is said to have flipped when the absolute value of its pitch angle exceeds 90 degrees.

- **Range-North** (for Ant environment): There is a safe area defined as a circle with center at $(0, 5)$ and radius $r = 20$. A state is considered bad if the agent is outside this safe area. Additionally, if the agent is in the safe area and the location $(x, y)$ satisfies $y \geq |x|$, then the state is considered good.

- **Hole-North** (for Ant environment): There are four holes in the scene, defined as circles with centers at $(15, 15)$, $(15, -15)$, $(-15, -15)$, $(-15, 15)$, and a common radius $r = 12$. A state is considered bad if the agent is in either hole. Additionally, if the agent is not in any of the holes and the location $(x, y)$ satisfies $y \geq |x|$, then the state is considered good.

**Metrics.** We employ three main metrics for evaluation:

- **Safe state coverage**, which measures the agent's ability to explore the state space while avoiding hazardous regions. Following (Kim et al., 2024), this metric assigns a value $+1, -1$ to safe and unsafe areas, and computes state coverage by counting the unique $1 \times 1$ x-y bins (or 1-unit x-axis bins for HalfCheetah tasks) visited by the agent. For Safety-Gym tasks, we set the x-y bin size to $0.01 \times 0.01$ due to the small scale of the coordinates.

- **Safe state ratio**, which quantifies the proportion of visited safe bins among all visited bins. It is defined as the ratio of the number of unique safe bins (labeled as good or neutral) to the number of all visited bins.

- **Downstream task performance**, which evaluates the utility of learned skills in downstream tasks. We consider both a zero-shot setting and a task-specific hierarchical control setting. To assess zero-shot performance, we roll out the downstream task environment with randomly sampled skills and report both the average and the best performance across all sampled skills. To assess hierarchical control performance, we use the learned skills as a low-level controller and train an additional high-level controller to optimize performance on the downstream task. The downstream task performance is then reported as the hierarchical control performance of the pretrained skills. Further details on the downstream tasks can be found in Appendix G.2, while additional information about the high-level controller is provided in Appendix G.3.

### G.2. Downstream Task Details

**Zero-shot performance.** We evaluate the zero-shot performance of the learned skills (in Ant tasks) on a customized Ant motion task. The single-step reward comprises a survival reward (1.0 per step), a movement reward (the maximum of the forward velocity and the lateral velocity), and a safety penalty ($-20.0$ if the agent enters the unsafe area defined by the guidance task used during skill learning).

**Hierarchical control performance.**   We evaluate the hierarchical control performance of the learned skills (in HalfCheetah Not-flip tasks) on a HalfCheetah Goal task. The agent will receive a reward of $1.0$ if it is sufficiently close to the goal (i.e., within a distance of less than 3), and a safety penalty of $-20.0$ if the agent flips. The goals are randomly sampled from the range $[-100, 100]$.

### G.3. Implementation Details

We implement METRA on top of the publicly available CSF codebase[1] (Zheng et al., 2025), as it provides more detailed scripts and supports the evaluation of downstream task performance. Code is available at `https://github.com/iii iiii11/COLLIE`.

For the baselines, we adopt the implementation of METRA (Park et al., 2024), DIAYN (Eysenbach et al., 2019), and LSD (Park et al., 2022b) from the CSF codebase, and implement online variants of DoDont (DoDont*) and DDG (DDG*) on top of the CSF codebase.

For DoDont*, to adapt the original DoDont (Kim et al., 2024) to our setting, we make two necessary modifications. First, since no offline datasets are available, the dataset for classifier training is collected along the skill discovery process. Specifically, every 1000 (state-based) or 1500 (pixel-based) epochs, we sample two trajectories from the replay buffer to construct the dataset. The dataset is of the same scale as the original DoDont. Second, as the dataset must be collected gradually, the classifier can only be trained incrementally. Specifically, each time the feedback buffer is updated, we fine-tune the classifier using the accumulated dataset collected up to that point. For DDG*, we apply the same modifications. Additionally, since DDG is built on DIAYN, which significantly underperforms METRA, we implement DDG* on top of METRA for a cleaner comparison of its guidance mechanism.

COLLIE shares the same hyperparameters as the baselines, which are consistent with METRA. We list these hyperparameters in Table 14.

*Table 14.* Common hyperparameters for unsupervised skill discovery methods.

| Hyperparameter | Value |
| --- | --- |
| Encoder for pixel tasks | CNN (LeCun et al., 1989) |
| # hidden layers | 2 |
| # hidden units per layer | 1024 |
| Learning rate | 0.0001 |
| Optimizer | Adam (Kingma & Ba, 2015) |
| Minibatch size | 256 |
| Target network smoothing coefficient | 0.995 |
| Entropy coefficient | auto-adjust (Haarnoja et al., 2018) |
| Total horizon length | 200 |
| # epoches | 5000 (Quadruped, Humanoid), 10000 (Ant, HalfCheetah, Safety-Gym) |
| # episodes per epoch | 8 |
| # gradient steps per epoch | 200 (Quadruped, Humanoid), 50 (Ant, HalfCheetah, Safety-Gym) |
| Discount factor $\gamma$ | 0.99 |
| METRA $\epsilon$ | $10^{-3}$ |
| METRA initial $\lambda$ | 30 |

---

[1] https://github.com/Princeton-RL/contrastive-successor-features

The additional hyperparameters of COLLIE are listed in Table 15.

*Table 15.* Additional hyperparameters for COLLIE.

| Hyperparameter | Value |
|---|---|
| Segment length $H$ | 20 |
| Feedback frequency $K$ | 1000 |
| Warm-up epochs before the first feedback | 2000 |
| The total feedback amount $N_{\text{total}}$ | 40 (Ant, HalfCheetah, Safety-Gym) |
| | 100 (Quadruped, Humanoid) |
| The feedback amount per session $M$ | 5 |
| Smoothing factor $k_\beta$ | 5 |
| Number of candidate queries $N_{\text{c}}$ | 50 |
| Number of nearest neighbors $k$ | 5 |

For hierarchical control tasks, we use a PPO (Schulman et al., 2017) agent as the high-level controller. The trained skills serve as the low-level controller, and their parameters are fixed during the training of the hierarchical control agent. The hyperparameters are shown in Table 16.

*Table 16.* Hyperparameters for high-level controllers.

| Hyperparameter | Value | Hyperparameter | Value |
|---|---|---|---|
| Learning rate | 0.0001 | Option timesteps length | 25 |
| Total horizon length | 200 | Replay buffer batch size | 256 |
| # hidden layers | 2 | # hidden units per layer | 1024 |
| Temperature $\alpha$ | 1 | # epochs | 9000 |
| # episodes per epoch | 8 | | |

