# OpenReview forum: "COLLIE: Guiding Skill Discovery in Semantically Coherent Latent Space"
_ICML.cc/2026/Conference — ICML 2026 regular_

### Official Review · Reviewer_sRqd · 2026-03-06

**Soundness:** 3
**Presentation:** 4
**Significance:** 2
**Originality:** 2
**Overall Recommendation:** 3
**Confidence:** 3

**Summary:**

This paper proposes COLLIE, a novel framework for Guided Skill Discovery (GSD) in reinforcement learning. Unlike traditional Unsupervised Skill Discovery (USD), which often suffers from task-irrelevant exploration, GSD incorporates guidance from a human labeler to steer the agent toward semantically meaningful regions of the state space. Specifically, COLLIE utilizes feedback (“good,” “neutral,” and “bad”) to construct a semantically coherent latent space. By integrating this guidance into a distance-maximizing skill discovery (DSD) objective, the method modulates the Lipschitz constraint on state embeddings: states near human-preferred examples receive a larger relaxation of the distance constraint, thereby incentivizing exploration within those regions. Empirical evaluations across standard RL benchmarks demonstrate that COLLIE achieves superior performance and safety compared to existing GSD baselines, such as a modified version of DoDont.

**Compliance With Llm Reviewing Policy:**

Affirmed.

**Final Justification:**

While I have increased my score following the rebuttal, I remain somewhat skeptical regarding the algorithmic novelty in this specific setting and the choice of baselines.

**Key Questions For Authors:**

### Questions for the Authors

* **DoDont Performance:** On page 7, the paper states: *“In contrast, DoDont*, which depends on a trained instruction network, performs worse, likely due to the network’s instability with limited feedback data.”* Could you elaborate on the specific nature of this instability? Specifically, does the modified version of DoDont attempt to train a classifier on a dataset too small to achieve reliable generalization, thereby leading to poor reward signals?

* **Methodological Complexity:** Given that the method already utilizes human-labeled data, why is the proposed integrated objective preferred over a simpler, split reward structure? For example, one could use a dual-term reward function consisting of (1) a standard unsupervised skill discovery term and (2) a guided exploration bonus derived from the $L_2$ distance to the nearest labeled states (essentially using the $w(s)$ logic as an additive reward).

**Limitations:**

A formal limitations section is missing. Please add a brief paragraph summarizing the study's assumptions and discussing any inherent limitations in the proposed method or evaluation process.

**Strengths And Weaknesses:**

### Strengths
* **Clarity:** The paper is exceptionally well-written, with a clear methodology and accessible core ideas.
* **Relevance:** Guided Skill Discovery (GSD) is a timely topic that effectively bridges unsupervised discovery and task-specific guidance using sparse human feedback.
* **Performance:** COLLIE consistently outperforms all chosen baselines across the evaluated environments.

### Weaknesses
* **Biased Baseline Selection:** Comparing COLLIE primarily against Unsupervised Skill Discovery (USD) methods is an imbalanced comparison, as USD lacks human labels. With only one GSD baseline (modified DoDont), it is difficult to isolate the true benefits of the COLLIE framework. More GSD baselines are needed.
* **Technical Novelty:** The methodology is closely related to the DoDont framework. The primary distinction—deriving $w(s)$ from latent distances instead of classifier scores—may be an incremental contribution, particularly as performance gaps are often narrow.
* **Missing Implementation Details:** The paper lacks detail on how the DoDont baseline was modified (absent from both text and appendix). A more rigorous discussion of the exact differences is required to understand what drives COLLIE's improvements.
* **Coherence Assumption:** The assumption that adjacent states share similar desirability may fail in "cliff-edge" environments, where a single transition can lead from a safe state to an irreversible "bad" state.

---

> ### Author Rebuttal · Authors · 2026-03-31
>
> Dear Reviewer,
>
> We sincerely thank you for your insightful feedback. Below, we address each of your concerns.
>
>
> **W1: More GSD baselines**
>
> **A for W1:**
> Thanks for the valuable comment.
> As suggested, we added an additional GSD baseline: DDG [1]. We adapt it to our setting by using labeled good/neutral states as positive samples and replay buffer samples as negatives, with the same query method as COLLIE. Since DDG is built on DIAYN, which may limit its performance, we also implement a METRA-based variant for fair comparison, denoted DDG-METRA.
>
> Results show that DDG performs poorly. DDG-METRA improves but still underperforms DoDont∗ and COLLIE. We attribute this to DDG’s feature‑engineering approach, which cannot be integrated into the DSD framework as a distance metric.
>
> |Method|Ant North|Ant Range|Ant Hole|
> |-|-|-|-|
> |DDG|-4±1|4±1|4±1|
> |DDG-METRA|-743±821|-585±335|487±275|
> |METRA|-1425±756|-1247±147|1179±147|
> |DoDont∗|1307±188|-427±224|1132±171|
> |COLLIE|1333±129|362±94|1149±127|
>
> [1] Learning Task Agnostic Skills with Data-driven Guidance (2021)
>
> **W2: Novelty and Performance gaps**
>
> **A for W2:**
>
> **Performance gap.** As shown in Table 1 in the paper, COLLIE consistently outperforms DoDont∗ across all tasks, with significant gaps in tasks like Ant Range (362±95 vs. -428±224).
> Also, DoDont∗ shows much higher variance, indicating instability under sparse, cumulatively collected feedback.
>
> **Novelty.** Our core contribution is not only replacing classifier scores with latent distances, but showing that training‑free geometric propagation is more robust than learning a classifier in sparse‑feedback scenarios.
> Specifically:
> 1. COLLIE requires no pre-collected expert demonstrations, unlike DoDont.
> 2. We explicitly connect temporal adjacency to semantic coherence via Eq.5, justifying the usage of temporal distance in the GSD framework. We believe this complements GSD methods like DoDont.
> 3. The training-free design eliminates the need for an auxiliary network and avoids overfitting risks, making COLLIE suitable for real-world settings with only sparse feedback.
>
> **W3: Details of DoDont∗**
>
> **A for W3:**
> The main modifications include:
> 1. Online data collection. Two trajectories are sampled from the replay buffer every 1000(state-based)/1500(pixel-based) epochs to construct the dataset. The dataset is of the same scale as the original DoDont.
> 2. Periodic fine-tuning. Each time the feedback buffer is updated, the classifier is fine-tuned using the accumulated dataset collected up to that point.
>
> These modifications are necessary to adapt DoDont to our setting.
>
> **W4: Coherence in cliff-edge scenarios**
>
> **A for W4:**
> Thanks for your comment.
> We would like to clarify that our assumption, i.e., $P[g(s)=g(s')]≥1-ε, ∃ε>0, ∀(s, s')∈S_{adj}$, is probabilistic, which can be understood as a trajectory-level coherence: along a typical trajectory, human desirability does not change erratically.
> In the cliff‑edge example, preference changes only at the moment of falling, but remains coherent before and after.
> This aligns with our assumption, as we do not require all adjacent state pairs to share labels, only that such transitions are rare enough not to disrupt the latent space structure.
>
> **Q1: Why DoDont performs poorly**
>
> **A for Q1:**
> The reason is twofold:
> 1. At the beginning of training, the feedback buffer contains only a small number of segments, which requires the classifier to be trained on extremely limited data. This could cause overfitting.
> 2. Training samples are collected online from the replay buffer, which contains trajectories from early, undirected policies. These trajectories may contain misleading patterns for the classifier, exacerbating overfitting.
>
> COLLIE avoids this by using a training-free geometric signal from the well-structured latent space.
>
> **Q2: Performance with split reward structure**
>
> **A for Q2:**
> Thanks for your comment.
> Our reward is derived by integrating the guidance signal as a distance constraint within the DSD framework, following the theoretical derivation in Sections 3.2-3.3, which is not a heuristic.
>
> As suggested, we conducted an ablation to test the additive alternative $r=(Φ(s')-Φ(s))^Tz+w(s)$.
> The additive reward degrades performance, confirming that our multiplicative formulation better preserves the theoretical grounding in Sections 3.2-3.3.
>
> ||Ant North|Ant Range|Ant Hole|
> |-|-|-|-|
> |COLLIE|1333±129|362±94|1149±127|
> |Additive|941±334|314±225|450±274|
>
> **L1: Limitation**
>
> **A for L1:**
> One limitation of COLLIE is the dead zone phenomenon (Appendix E): in environments with obstacles, like Ant Hole, the learned skills may fail to explore regions hidden behind obstacles.
> COLLIE inherits this issue from METRA: the DSD objective aligns skills with the dominant geometry, and without explicit exploration bonuses, regions behind obstacles may remain underexplored.
> Addressing this by combining DSD with exploration mechanisms is a promising future direction.

---

> > ### Author Rebuttal · Reviewer_sRqd · 2026-04-03
> >
> > Thank you for the rebuttal and the additional experiments. While the inclusion of new baselines is helpful, the technical novelty relative to this specific problem domain remains a concern. The comparison with existing GSD methods is still somewhat limited, and it remains difficult to determine if the modifications to the DSG baselines ensure a fair evaluation. Specifically, the advantage of using an active human labeler during skill acquisition—compared to methods like DoDont that rely on static offline data—complicates a direct benchmark comparison.
> >
> >
> > # Further Comments:
> >
> >
> > ## Novelty and contribution
> >
> > I find the technical contributions to be marginal. The method is still very similar to DoDont. Overall, COLLIE introduces a human labeler and derives the guidance signal from latent space distances shaped by human preferences instead of using a classifier directly.
> >
> > > COLLIE requires no pre-collected expert demonstrations, unlike DoDont.
> >
> > While DoDont (from https://arxiv.org/pdf/2406.00324) relies on precollected data, COLLIE leverages a human labeler in the loop. I find this to be a much stronger assumption and more costly to implement in real world applications compared to using offline datasets.
> >
> > > The training-free design eliminates the need for an auxiliary network and avoids overfitting risks, ..
> >
> > I find the claim that a training free design is a primary benefit to be unconvincing. The main bottleneck for real world use is the reliance on a human in the loop rather than the computational requirement of training an additional network. Overfitting risks with limited data can be mitigated through other established means such as finetuning or regularization.
> >
> > > the training-free design eliminates the need for an auxiliary network and avoids overfitting risks, making COLLIE suitable for real-world settings with only sparse feedback.
> >
> > As written above, I find that the reliance on a human in the loop is rather a drawback and limits real-world use since human labelers are slow and costly.
> >
> >
> > ## Label coherence across trajectory
> >
> > Thank you for your response. I acknowledge that this assumption is probabilistic rather than hard. Nevertheless, the method still assumes that non-coherent cases occur only rarely. In practice, especially within long horizon trajectories, the quality of a segment might consist of a mix of behaviors, such as moving toward a goal while simultaneously getting stuck or approaching a hazardous area. In these instances, a single label for the entire trajectory segment remains an imprecise signal.

---

> > > ### Author Response · Authors · 2026-04-04
> > >
> > > Dear Reviewer,
> > >
> > > Thanks for the detailed comments.
> > >
> > > We must point out that **our method and DoDont [1] target different settings and should not be compared as equivalents**.
> > > COLLIE operates in the online guidance setting (human feedback available, no expert data), while DoDont assumes offline expert data.
> > > This distinction mirrors that between preference-based RL and offline RL.
> > > They address different scenarios, each with its own assumptions and applicability, and thus should not be compared directly.
> > >
> > > **Limited comparison with existing GSD methods**
> > >
> > > We have added a new GSD baseline as you suggested.
> > > We note that existing GSD methods are limited in number, and many rely on expert demonstrations or pre-defined constraints that are incompatible with our online, sparse-feedback setting.
> > > If you have a specific GSD method in mind that aligns with this setting, please provide the specific citation.
> > > We would be glad to address it.
> > >
> > > **Fairness of the comparison with DoDont [1]**
> > >
> > > As discussed above, COLLIE and DoDont focus on different settings (online human feedback vs. offline expert data), making direct comparison infeasible without adaptation.
> > > We have tried our best to ensure a fair comparison.
> > > Our modifications to DoDont were minimal and strictly necessary to enable evaluation under our setting, as detailed in the rebuttal.
> > > If any specific adaptation raises concerns about fairness, we welcome concrete suggestions to address them directly.
> > >
> > > **Novelty and contribution**
> > >
> > > **1. Similarity with DoDont**
> > >
> > > We respectfully disagree that similarity in algorithmic form implies incremental contribution.
> > > Many GSD methods [2-3] can be expressed in a similar formulation (Sec 2), yet are not merely incremental.
> > > We note that COLLIE is derived step-by-step from the online guidance setting and introduces an active query selection mechanism, which is not present in DoDont.
> > > Moreover, we identify a common limitation of prior GSD methods: they use only limited expert data for guidance, ignoring abundant unsupervised data from skill discovery.
> > > COLLIE leverages this data to construct a training-free guidance signal, eliminating overfitting risk and yielding significant performance gains.
> > >
> > > **2. Assumption on human labeler**
> > >
> > > We respectfully disagree that a human labeler is a stronger assumption than offline expert data.
> > > This view implicitly assumes that high-quality expert demonstrations are readily available, yet curating such datasets requires significant expert time and domain knowledge, making them costly [4-5].
> > > Also, in many realistic scenarios like robotic arm operation, non-expert users can easily provide "good/bad" labels.
> > > Thus, obtaining high-quality offline data (as DoDont requires) is not necessarily easier than collecting human labels, and the human-in-the-loop setting is meaningful and practically relevant.
> > >
> > > We respectfully note that the reviewer's criticisms primarily target our problem setting rather than our contribution.
> > > We kindly ask the reviewer to evaluate the contribution within its intended setting, rather than dismissing it based on a different setting.
> > >
> > > **3. Alternate method to mitigate overfitting**
> > >
> > > Fine-tuning or regularization can mitigate overfitting, but introducing additional hyperparameters and complexity without fully resolving it.
> > > In this paper, we identify the root cause: prior works rely solely on limited expert data and ignore unsupervised skill discovery data.
> > > COLLIE exploits this unsupervised data for guidance construction, requires no auxiliary network, and thus completely avoids overfitting.
> > >
> > > **Label coherence across trajectory**
> > >
> > > We would like to note that the issue you raised relates to segment length rather than our coherence assumption (nearby states share similar desirability).
> > >
> > > For the example "moving toward a goal while simultaneously getting stuck or approaching a hazardous area", our labeling protocol explicitly addresses this: "We label a segment as bad if it contains any undesirable state" (Sec. 4.1), reflecting a conservative safety preference.
> > > This conservative labeling reflects human preferences for safety.
> > > Empirically, COLLIE remains robust under varying segment lengths (Table 12) and noisy labels (Table 5).
> > > Furthermore, DoDont would also label such a segment as "Don't" without resolving ambiguity, and its apparent robustness likely stems from offline data excluding such mixed behaviors.
> > >
> > > Thanks again for the detailed comments. We sincerely hope that you could re-evaluate our work after reading our response.
> > >
> > > Best,
> > >
> > > The Authors
> > >
> > > [1] Do's and Don'ts: Learning Desirable Skills with Instruction Videos. NeurIPS 2024.
> > >
> > > [2] Safety-aware unsupervised skill discovery, ICRA 2023.
> > >
> > > [3] Human-Aligned Skill Discovery: Balancing Behaviour Exploration and Alignment. AAMAS 2025.
> > >
> > > [4] RoboCLIP: One Demonstration is Enough to Learn Robot Policies. NeurIPS 2023.
> > >
> > > [5] A Survey on Imitation Learning for Contact-Rich Tasks in Robotics. The International Journal of Robotics Research, 2026.

---

### Official Review · Reviewer_NFHC · 2026-03-11

**Soundness:** 2
**Presentation:** 2
**Significance:** 2
**Originality:** 3
**Overall Recommendation:** 3
**Confidence:** 3

**Summary:**

This paper proposes COLLIE, a GSD framework that constructs a semantically coherent skill latent space to efficiently utilize sparse human feedback. By doing so, it aims to overcome limitations of existing GSD methods that either rely on pre-defined rules or expert demonstrations or require training auxiliary models to encode human intent.

**Compliance With Llm Reviewing Policy:**

Affirmed.

**Final Justification:**

I appreciate the authors’ detailed response to my comments. However, the experimental setup used for the comparison with DoDont in the Kitchen environment still leaves me with concerns about the scalability of COLLIE, as I originally noted in W1. Training DoDont relies on generic datasets, such as D4RL datasets together with random datasets, rather than datasets specifically tailored to a particular task. In contrast, COLLIE appears to require an additional process of distinguishing between good and bad states. This may require more human involvement, and such distinctions could become increasingly ambiguous in more complex environments. In the experiments presented by the authors, state labeling was conducted based on a clear distinction between good tasks and bad tasks. However, because of this relatively explicit criterion, it remains difficult to assess whether COLLIE can overcome the scalability limitation arising from the ambiguity of state labeling. For this reason, I am not yet convinced that the current results sufficiently address my concern regarding scalability. Therefore, I will maintain my original score (weak reject).

**Key Questions For Authors:**

1. Aside from the difference in how guidance is provided (video examples vs. human intervention), another key distinction seems to be that DoDont labels transitions, whereas COLLIE labels states. As a result, DoDont may better capture behavior-level intentions, while COLLIE might struggle to reflect such intentions. Could the authors comment on this difference?
2. In COLLIE, guidance is defined over states as (w(s)). Is it possible to extend the proposed approach to transition-level guidance (w(s, s'))?
3. From the same state, different actions may lead to either desirable or undesirable states. However, COLLIE assumes that states that are close in the latent space share similar human desirability. Could this assumption lead the model to learn overly conservative skills?
4. Including results from the Kitchen environment used in the DoDont experiments could further support the claim regarding the scalability and generalizability of COLLIE.

**Limitations:**

The paper mentions potential negative societal impacts, but it does not discuss the limitations of the proposed framework.

**Strengths And Weaknesses:**

**Strengths**

- COLLIE enables effective guidance with only a small amount of labeled data, without requiring labels or rewards for all states or actions in the dataset.
- The training-free guidance signal used in COLLIE allows skill discovery to be trained efficiently.

**Weaknesses**

- There are concerns regarding scalability to more diverse tasks. COLLIE trains the agent to avoid visiting undesirable states by distinguishing between good and bad states. While this approach may work well for navigation tasks such as those involving Ant or Humanoid, it may face limitations in manipulation tasks such as Kitchen, which require more complex movements and richer state representations.
- According to the active query strategy described in Section 3.4, COLLIE prioritizes querying states that are less visited in the labeled dataset. While this strategy may help improve coverage as claimed in the paper, it does not necessarily guarantee informative queries. For example, querying states near the decision boundary between good and bad states may be more informative, especially for complex tasks.
- The paper does not explicitly discuss the limitations of COLLIE.

---

> ### Author Rebuttal · Authors · 2026-03-31
>
> Dear Reviewer,
>
> We sincerely thank you for your insightful and constructive feedback.
> We are grateful for your valuable comments, which have helped us enhance the quality of our manuscript.
> We hope the following statement clears your concern.
>
> **W1, Q4: Applicability in manipulation tasks such as the Kitchen**
>
> **A for W1, Q4:**
> Thanks for your comment.
> While our pixel-based Quadruped and Humanoid experiments already involve rich state representations, we agree that evaluation on manipulation tasks further strengthens the claim of generalizability.
> As suggested, we conducted additional experiments in the Kitchen environment.
> - **Task design.** We define three "good" tasks (bottom_burner_knob, light_switch, slide_cabinet_handle) and three "bad" tasks (hinge_cabinet_handle, microwave_handle, kettle_top). A state is labeled as good only if any good tasks are completed and no bad tasks are completed; a state is labeled as bad if any bad tasks are completed.
> - **Evaluation.** We measure the average number of good tasks completed per trajectory (counting a task as completed if it appears at any state in the trajectory).
> Due to limited computational resources, all algorithms are trained for 1,500 epochs. Query-related parameters are consistent with other pixel tasks, and other parameters are consistent with METRA.
> - **Results.** As shown in the table below, COLLIE outperforms DoDont∗ in this challenging manipulation setting, confirming that our method is not narrowly confined to locomotion tasks.
>
> |Method|Kitchen|
> |-|-|
> |Oracle|0.94 ± 0.01|
> |DoDont*|0.31 ± 0.07|
> |COLLIE|0.42 ± 0.10|
>
> **W2: Query strategy does not necessarily guarantee informative queries**
>
> **A for W2:**
> Thanks for your comment.
> As suggested, we conduct an ablation using the suggested query selection method ("Uncertainty"), which queries states near the decision boundary between good and bad states, prioritizing states with large guidance signal entropy: $H_w(s)=H[\text{softmax}([-d_Φ(s, D_i)]_{i=0}^2)]$.
> As shown in the table below, COLLIE consistently outperforms Uncertainty across all tasks, for it prioritizes under-explored regions, ensuring effective guidance across a wider state space under sparse feedback.
> These results underscore the critical role of exploration in query selection.
>
> |Method|Ant Hole|Ant North|Ant Range|
> |-|-|-|-|
> |COLLIE|1149.20 ± 127.05|1333.20 ± 129.10|329.80 ± 110.41|
> |Uncertainty|1034.40 ± 468.49|1059.20 ± 79.16|-79.60 ± 302.86|
>
>
> **W3, L1: Limitation**
>
> **A for W3, L1:**
> One limitation of COLLIE is the "dead zone" phenomenon in environments with obstacles.
> As detailed in Appendix E, COLLIE inherits this issue from the underlying METRA framework. In tasks like Ant Hole, learned skills may fail to explore regions hidden behind obstacles.
> This occurs because the DSD objective encourages skills to align with the dominant geometry of the accessible state space instead of maximizing state coverage, leading to movement along open, direct paths in the state space.
> Thus, without explicit exploration mechanisms such as prediction error or state entropy, these obscured regions become underexplored.
> Future work combining DSD with explicit exploration methods is a promising direction for addressing this issue.
>
> **Q1-2: State-level vs. transition-level guidance**
>
> **A for Q1-2:**
> Thanks for the insightful questions.
> Our state-level guidance w(s) is motivated by two practical considerations under sparse feedback:
> 1. The state space is substantially smaller than the transition space. Reducing the solution space mitigates overfitting with limited labels.
> 2. State labels enable direct propagation via nearest-neighbor distances in the DSD-constructed latent space Φ(s), whereas transition labels would require modeling consecutive states, which would likely necessitate an auxiliary model and additional training, introducing overfitting risks under sparse feedback.
>
> Extending to transition-level guidance is theoretically feasible by substituting transitions (s,s') for states s in Eq.4-5, yet it faces the same practical challenges.
> Modifying the DSD framework to accept transitions as Φ inputs could partially address these, but it lies beyond this work's scope.
>
> **Q3: Reasonability of assumptions**
>
> **A for Q3:**
> Thanks for the insightful question.
> As you commented, the consequence of an action is reflected in the subsequent state s', meaning state labels implicitly capture action outcomes.
> A bad action leads to a state with low guidance values, so state-level guidance can effectively direct actions: an action is encouraged if it reaches a desirable state and discouraged otherwise.
> Thus, state-level guidance should be as effective as state-action-level guidance and won't lead to overly conservative skills.
> This aligns with the empirical results, where COLLIE achieves diverse skill coverage without sacrificing safety, validating our design choice.

---

> > ### Author Rebuttal · Reviewer_NFHC · 2026-04-02
> >
> > Thank you for the response. Most of my questions seem to have been addressed. However, I still have a follow-up question regarding the Kitchen experiment. In the DoDont paper, the Kitchen evaluation measures policy task coverage over six tasks, using the D4RL dataset as the Do data and random action videos as the Don’t data. Under that setup, DoDont is reported to achieve coverage of roughly five tasks.
> >
> > By contrast, the Kitchen experiment presented in the authors’ response to W1 and Q4 appears to differ from the original DoDont setup in both task design and evaluation protocol. In this regard, I would appreciate clarification on why the authors chose a different experimental design for Kitchen, and why DoDont exhibits such low performance under this setting compared to what is reported in the original paper. A clearer explanation of these differences would make it easier to interpret the newly added Kitchen results.

---

> > > ### Author Response · Authors · 2026-04-02
> > >
> > > Dear Reviewer,
> > >
> > > We are glad that our previous response addressed most of your concerns.
> > > We also appreciate the valuable comments, which helped us significantly enhance the quality of our manuscript.
> > > We hope the following statement clears your remaining concern.
> > >
> > >
> > > **Explanation of the different experimental design for Kitchen and the current DoDont performance:**
> > >
> > > Our setting differs fundamentally from the original DoDont paper.
> > > As you commented, DoDont trains its guidance using pre-collected expert demonstrations (D4RL) as "Do" data and random videos as "Don't" data.
> > > However, in this work, we focus on an "online" setting, where the guidance is learned from trajectories sampled by the evolving policy, and no pre‑collected dataset is available.
> > > Consequently, a different experimental design is necessitated, which we detailed below.
> > >
> > > 1. **Different choice of positive/negative samples.**
> > >
> > > As mentioned above, using D4RL as positive samples is infeasible in our online setting, for we cannot access pre-collected datasets.
> > > Also, using random trajectories as negative samples, as in DoDont, would hinder the early exploration of skills.
> > > Since most initial trajectories are nearly random, labeling them as "bad" would drive intrinsic rewards near zero early in training.
> > > Therefore, we designed a task‑based scheme: states that complete a subset of tasks are labeled "good", and states that complete another subset are labeled "bad".
> > > This provides clear guidance without suppressing early exploration.
> > >
> > > 2. **Difference in evaluation protocol \& Explanation of current DoDont performance.**
> > >
> > > The performance discrepancy stems from different evaluation metrics.
> > >
> > > Our primary metric (COLLIE 0.42, DoDont∗ 0.31) measures the expected number of good tasks completed by a *single* skill.
> > > It measures the independent usability of individual skills, serving as a more fine-grained, discriminative metric than aggregate coverage in DoDont.
> > > Since a single skill rarely completes multiple tasks, this metric naturally stays low (≤1).
> > > This makes it incomparable to DoDont's original task coverage metric, which counts tasks covered by *any* skill across the whole skill set.
> > >
> > > To ensure a fair comparison, we additionally report "Good task coverage", i.e., the number of good tasks completed by at least one skill. Since we defined three good tasks, the maximum coverage is 3.
> > > As shown below, both DoDont∗ and COLLIE achieve coverage of 2 out of 3 good tasks, which is consistent with DoDont's original reported coverage (approximately 4–5 out of 6 tasks).
> > >
> > > | Method | Our primary metric | Good task coverage (max=3) |
> > > |--|--|--|
> > > |DoDont∗| 0.31 ± 0.07 | 2.00 ± 0.00|
> > > |COLLIE| 0.42 ± 0.10 | 2.00 ± 0.00|
> > >
> > > We sincerely thank the reviewer again for the thoughtful and constructive feedback.
> > > We hope that our responses have addressed your concerns.

---

### Official Review · Reviewer_jL5Y · 2026-03-12

**Soundness:** 3
**Presentation:** 4
**Significance:** 3
**Originality:** 3
**Overall Recommendation:** 5
**Confidence:** 4

**Summary:**

The paper proposes COLLIE, a framework for guided skill discovery that incorporates sparse human feedback into unsupervised skill discovery. The main idea is to learn a latent representation of states that is semantically coherent, such that nearby embeddings correspond to states with similar human desirability. Given a small set of labeled states (good, neutral, bad), the method constructs a dense guidance signal by propagating labels in the latent space using distances to labeled examples, avoiding the need to train an auxiliary reward or guidance model. This guidance signal is then incorporated into the intrinsic reward used for distance-based skill discovery. Experiments on several locomotion environments show that the method learns diverse skills while avoiding undesirable behaviours and improves downstream task performance compared to existing baselines.

**Compliance With Llm Reviewing Policy:**

Affirmed.

**Final Justification:**

I think the paper proposes something valuable to the field guided skill discovery. Authors show improved performance over baseline methods and justify their method appropriately.

**Key Questions For Authors:**

- Can you show that this latent guidance can be applied to other state spaces that are not continuous or more noisy?

**Limitations:**

The authors have adequately addressed limitations.

**Strengths And Weaknesses:**

Strengths:
- I think that adding the guidance signal as a constraint to the skill objective is clever and augments the skill discovery objective in a principled way.
- The idea of incorporating a guidance signal directly into the DSD objective is elegant and integrates human feedback into skill discovery in a principled way without requiring an auxiliary reward model.
- The training-free guidance construction is appealing in practice, as it avoids learning a separate reward or classifier from sparse feedback.
- The experimental evaluation is thorough across several locomotion environments and includes useful ablations (feedback sparsity, noise robustness, and latent-space coherence).

Weaknesses:
- I suspect a caveat with the approach of propagating the human signal across the latent is that this is mostly effective when the state-action space is coherent as well. The approach relies heavily on the assumption of a _semantically coherent latent space_ (Eq. 4), where nearby embeddings correspond to states with similar human desirability. While the authors motivate this assumption through temporal adjacency and demonstrate it in locomotion environments, it is unclear how robust this assumption would be in more complex or discontinuous state spaces. In tasks involving abstract decision sequences or multimodal dynamics, states that are close in the learned embedding may correspond to qualitatively different outcomes (e.g., safe vs. unsafe behaviors). In such cases, propagating sparse labels via nearest-neighbor distances in the latent space could lead to incorrect guidance signals. The paper would benefit from a discussion or evaluation of how sensitive COLLIE is to violations of this semantic coherence assumption.

- Conceptually, COLLIE largely builds on the existing Distance-maximizing Skill Discovery (DSD/METRA) framework and modifies the intrinsic reward using a guidance weight derived from nearest-neighbor distances to labeled states. While the resulting framework is elegant and empirically effective, the methodological novelty relative to recent GSD methods may be somewhat limited. In particular, the main novelty lies in the training-free construction of the guidance signal, which I feel applies quite narrowly to the robotic locomotion benchmarks.

---

> ### Author Rebuttal · Authors · 2026-03-31
>
> Dear Reviewer,
>
> We sincerely thank you for your insightful and constructive feedback.
> We are particularly encouraged by your positive assessment and recognition of the value of our work.
> We are grateful for your valuable comments, which have helped us enhance the quality of our manuscript.
> We hope the following statement clears your concern.
>
>
> **W1: Robustness of assumptions in complex scenarios**
>
> **A for W1:**
> Thank you for this thoughtful comment. We would like to clarify that Eq.4 describes a target property of our latent space, rather than an assumption.
>
> Directly enforcing this property is challenging. Thus, we leverage a more fundamental observation: human desirability does not change arbitrarily along trajectories. That is, adjacent states share the same label with high probability. We therefore impose a surrogate constraint: for adjacent states sampled from trajectories, their embeddings are kept close (Eq.5). This constructs a latent space that approximates the desired coherence.
>
> Importantly, our method makes no assumption about the structure of the state space itself. We only require human desirability to be locally consistent along trajectories. We believe this weaker, more natural assumption holds across a wide range of domains, including those involving abstract decision sequences or multimodal dynamics.
>
> In some extreme scenarios, if the trajectory‑level consistency is severely violated (e.g., desirability fluctuates rapidly along a trajectory), the latent space may fail to achieve semantic coherence, and COLLIE would degrade to METRA.
> This limitation is common to existing GSD methods, as in such extreme scenarios, every trajectory contains both good and bad behaviors.
> Consequently, deriving expert trajectories is also challenging for those methods.
>
>
> **W2.1: Methodological Novelty**
>
> **A for W2.1:**
> Thank you for the valuable comment.
> The core contribution of COLLIE lies not merely in the training-free construction of the guidance signal, but in (1) establishing a general online GSD pipeline without pre-collected high-quality expert data, and (2) providing theoretical analysis justifying the use of the DSD framework in guided settings. We believe these aspects apply beyond the robotic locomotion benchmarks.
> Specifically:
> 1. **No expert data required.** COLLIE requires no pre-collected expert demonstrations, unlike DoDont, which requires pre-collected expert trajectories for classifier training.
> 2. **Theoretical grounding.** We explicitly connect temporal adjacency to semantic coherence via Eq.5. This justifies using temporal distance in the GSD framework, an aspect not discussed in prior GSD works. We believe this also complements existing GSD methods built upon the DSD framework.
> 3. **Practical robustness.** The training-free design eliminates the need for an auxiliary network and avoids overfitting risks. This makes COLLIE suitable for real-world applications where expert data is unavailable and feedback is limited.
>
>
> **W2.2: Applicability beyond locomotion benchmarks**
>
> **A for W2.2:**
> Thank you for this valuable comment. To demonstrate that COLLIE's training-free guidance extends beyond locomotion benchmarks, we have conducted additional experiments on the Kitchen manipulation environment.
> - **Task design.** We define three "good" tasks (bottom_burner_knob, light_switch, slide_cabinet_handle) and three "bad" tasks (hinge_cabinet_handle, microwave_handle, kettle_top). A state is labeled as good only if any good tasks are completed and no bad tasks are completed. A state is labeled as bad if any bad tasks are completed.
> - **Evaluation.** We measure the average number of good tasks completed per trajectory (counting a task as completed if it appears at any state in the trajectory).
> Due to limited computational resources, all algorithms are trained for 1,500 epochs. Query-related parameters are consistent with other pixel tasks, and other parameters are consistent with METRA.
> - **Results.** As shown in the table below, COLLIE outperforms DoDont∗ in this challenging manipulation setting, confirming that our method is not narrowly confined to locomotion tasks.
>
> | Method | Kitchen |
> | -- | -- |
> | Oracle | 0.94 ± 0.01 |
> | DoDont∗ | 0.31 ± 0.07 |
> | COLLIE | 0.42 ± 0.10 |
>
>
>
> **Q1: Applicability in noisy state spaces**
>
> **A for Q1:**
> Thank you for the valuable comment.
> To evaluate robustness to state observation noise, we add Gaussian noise to the observed state: for an original state $s$, the observed input is $(1+x)s$ where $x\sim N(0, 0.01 I)$. As shown in the table below, performance degrades only slightly, demonstrating that COLLIE remains effective under moderate observation noise.
>
> | Method | Ant Range | Ant Hole |
> | -- | -- | -- |
> | COLLIE | 362.20 ± 94.55 | 1149.20 ± 127.05 |
> | COLLIE-noise | 325.00 ± 54.59 | 1009.00 ± 97.30 |

---

> > ### Author Rebuttal · Reviewer_jL5Y · 2026-04-04
> >
> > The authors have added an additional experiment which I consider to be valuable. Some concerns remain such as tasks that don't require learning skills as with locomotion task, but I think the authors have done enough to warrant an acceptance of this paper.

---

> > > ### Author Response · Authors · 2026-04-04
> > >
> > > Dear Reviewer
> > >
> > > We would like to thank the reviewer for raising the score! We sincerely appreciate your time, valuable comments, and thoughtful engagement throughout the review process, which helped us significantly improve the paper's strengths. We will incorporate all feedback in the final version. We are grateful for your support and wish you all the best.

---

### Official Review · Reviewer_R2Wi · 2026-03-13

**Soundness:** 4
**Presentation:** 4
**Significance:** 3
**Originality:** 3
**Overall Recommendation:** 5
**Confidence:** 3

**Summary:**

Guided skill discovery (GSD) is a promising technique to allow agents to explore skills while avoiding hazardous regions, whereas existing methods require pre-defined rules or additional auxiliary models. This paper proposes a method named COLLIE that can construct semantically coherent skill latent spaces from limited human guidance. The proposed method constructs a latent space in which state variables are embedded so that near latent variables have similar human desirability. Policies are trained based on the acquired latent variables. Results demonstrated that the proposed method enabled agents to explore desirable regions better than existing methods.

**Compliance With Llm Reviewing Policy:**

Affirmed.

**Final Justification:**

This reviewer is satisfied with the rebuttal and maintains the score of Accept.

**Key Questions For Authors:**

1. Although the results supported the main arguments, this reviewer would also like to see a result showing how the acquired latent spaces are coherent. In other words, can we actually observe the phenomenon where state variables that are distant but share similar desirability are located close together in the latent space? If it is possible to provide an extreme case (where far-distant state variables converge in the latent space), it would be strong evidence for the arguments.
2. The latent space would have a trade-off between coherence and the Lipschitz condition with respect to temporal distance. The latter one works as regularization, while limiting the representation ability of the encoder $\phi$ when a large deformation is necessary to make the latent space coherent. Is it a possible limitation of the proposed method?

**Limitations:**

This paper describes the advantages of the proposed method well, but lacks descriptions of its limitations. Please clarify limitations that would be hard to address and incompatible trade-offs (Key Question 2, for example).

**Strengths And Weaknesses:**

# Soundness

The methodology sounds effective, and the results clearly support the effect of the proposed method. This paper also provided comprehensive ablation studies, which reveal how each technique of the proposed method contributes to the performance.

# Presentation

This paper is readable. The main arguments are clearly described. Symbols and notations are also clearly defined.

# Significance

Guiding learning from limited human feedback is an important issue for practical self-supervised exploration.

# Originality

The approach of constructing a latent space in which nearby points have similar human desirability sounds promising for encoding human feedback. This paper will provide new insights into how sparse and limited human feedback should be generalized.

---

> ### Author Rebuttal · Authors · 2026-03-31
>
> Dear Reviewer,
>
> We sincerely thank you for your insightful and constructive feedback.
> We are particularly encouraged by your positive assessment and recognition of the value of our work.
> We are grateful for your valuable comments, which have helped us enhance the quality of our manuscript.
> We hope the following statement clears your concern.
>
>
> **Q1: Evidence of the coherent latent space**
>
> **A for Q1:**
> Thank you for the valuable comment.
> As suggested, to demonstrate the semantic coherence property of the learned latent space, we visualize labeled states from the Ant North task in both the original state space and the corresponding latent space.
> We use ISOMAP to project both high-dimensional spaces to 2D.
> Please refer to Figure 1 in the [anonymous link](https://anonymous.4open.science/r/icml2026_rebuttal_imgs-F589/r1_visualization.pdf).
>
> We observe two key phenomena:
> 1. Figure 1(a): States with different human desirability but close in the original state space are mapped to distant points in the latent space. This observation is exactly aligned with the definition of semantic coherence (Eq. 4).
> 2. Figure 1(b): States with similar desirability that are far apart in the original state space can converge to nearby points in the latent space. This observation confirms that the latent space successfully captures human desirability beyond raw state distances.
>
>
> **Q2: Trade-off between coherence and the Lipschitz condition**
>
> **A for Q2:**
> Thank you for the valuable feedback.
> The semantical coherence is derived using the Lipschitz condition with respect to temporal distance.
> Therefore, the coherence and the Lipschitz are consistent and do not induce a trade-off.
>
> Specifically, the coherence property in Eq.4 means that states being close in the latent space has large probability of having the same desirability. While directly enforcing this property is challenging, we instead leverage a more fundamental observation: human desirability does not change arbitrarily along trajectories. That is, adjacent states share the same label with high probability.
> We therefore impose a surrogate constraint: for adjacent states sampled from trajectories, their embeddings are kept close (Eq.5). This constraint is then shown to be equivalent to the Lipschitz condition you mentioned, which measures the temporal distance in the state space.
>
> **L1: Limitations**
>
> **A for L1:**
> Thank you for the valuable feedback.
> One limitation of COLLIE is the "dead zone" phenomenon in environments with obstacles.
> As detailed in Appendix E, COLLIE inherits this issue from the underlying METRA framework. In tasks like Ant Hole, learned skills may fail to explore regions hidden behind obstacles.
> This occurs because the DSD objective encourages skills to align with the dominant geometry of the accessible state space instead of maximizing state coverage, leading to movement along open, direct paths in the state space.
> Thus, without explicit exploration mechanisms such as prediction error or state entropy, these obscured regions become underexplored.
> Future work combining DSD with explicit exploration methods is a promising direction for addressing this issue.

---

> > ### Author Rebuttal · Reviewer_R2Wi · 2026-04-04
> >
> > The authors appropriately addressed this reviewer's concerns. The authors may wish to consider adding these clarifications to the manuscript (or the appendices).

---

> > > ### Author Response · Authors · 2026-04-04
> > >
> > > Dear Reviewer
> > >
> > > We sincerely appreciate your time, valuable comments, and thoughtful engagement throughout the review process, which helped us significantly improve the paper's strengths. We will incorporate all feedback in the final version. We are grateful for your support and wish you all the best.

---

### Decision · Program_Chairs · 2026-04-30

**Decision:**

Accept (regular)

**Comment:**

The paper proposes COLLIE, a guided skill discovery framework that uses sparse human feedback to construct a semantically coherent latent space and derive a training-free guidance signal for skill learning. The paper addresses an important problem in guided exploration, and the idea of leveraging dense unsupervised data to avoid training an additional guidance model is both practically appealing and conceptually clean.

The reviewer discussion reflects some disagreement, but in my view the overall case for the paper is positive. Two reviewers were clearly supportive, highlighting the principled integration of human feedback, the practical appeal of the training-free guidance mechanism, and the strong empirical performance. The rebuttal also strengthened the paper by adding a new GSD baseline, providing additional experiments on Kitchen, and clarifying the intended online sparse-feedback setting relative to prior work such as DoDont. These additions make the contribution and scope of the paper clearer.

The main remaining concern is about scalability and generality beyond the evaluated domains, especially in settings where state-level good/bad labeling becomes more ambiguous. I think this is a fair limitation, and the current evidence does not fully settle it. However, I do not view it as sufficient to outweigh the strengths of the paper, particularly given that the method is evaluated in both state- and pixel-based settings and that the rebuttal provides useful additional evidence beyond locomotion alone.

Overall, I believe the paper makes a solid contribution to guided skill discovery, with a practically useful idea, strong motivation, and sufficiently convincing empirical support.